# Does Invariant Graph Learning via Environment Augmentation Learn Invariance?

**Yongqiang Chen**[1],[*] **Yatao Bian**[2], **Kaiwen Zhou**[1]
[1]The Chinese University of Hong Kong [2]Tencent AI Lab
{yqchen,kwzhou}@cse.cuhk.edu.hk yatao.bian@gmail.com

**Binghui Xie**[1], **Bo Han**[3], **James Cheng**[1]
[3]Hong Kong Baptist University
bhanml@comp.hkbu.edu.hk {bhxie21,jcheng}@cse.cuhk.edu.hk

## Abstract

Invariant graph representation learning aims to learn the invariance among data from different environments for out-of-distribution generalization on graphs. As the graph environment partitions are usually expensive to obtain, augmenting the environment information has become the *de facto* approach. However, the *usefulness* of the augmented environment information has never been verified. In this work, we find that it is fundamentally *impossible* to learn invariant graph representations via environment augmentation without additional assumptions. Therefore, we develop a set of *minimal assumptions*, including variation sufficiency and variation consistency, for feasible invariant graph learning. We then propose a new framework **G**raph inv**A**riant **L**earning **A**ssistant (`GALA`). `GALA` incorporates an assistant model that needs to be sensitive to graph environment changes or distribution shifts. The correctness of the proxy predictions by the assistant model hence can differentiate the variations in spurious subgraphs. We show that extracting the maximally invariant subgraph to the proxy predictions provably identifies the underlying invariant subgraph for successful OOD generalization under the established minimal assumptions. Extensive experiments on 12 datasets including DrugOOD with various graph distribution shifts confirm the effectiveness of `GALA`[1].

## 1 Introduction

Graph representation learning with graph neural networks (GNNs) has proven to be highly successful in tasks involving relational information [24, 35, 68, 74, 75]. However, it assumes that the training and test graphs are independently drawn from the identical distribution (iid.), which can hardly hold for many graph applications such as in Social Network, and Drug Discovery [25, 26, 37, 70, 86, 94]. The performance of GNNs could be seriously degenerated by *graph distribution shifts*, i.e., mismatches between the training and test graph distributions caused by some underlying environmental factors during the graph data collection process [16, 21, 28, 70, 87, 94]. To overcome the Out-of-Distribution (OOD) generalization failure, recently there has been a growing surge of interest in incorporating the invariance principle from causality [60] into GNNs [8, 17, 41, 45, 55, 72, 73, 77, 83]. The rationale of the invariant graph learning approaches is to identify the underlying *invariant subgraph* of the input graph, which shares an invariant correlation with the target labels across multiple graph

---

[*]Work done during an internship at Tencent AI Lab.

[1]Code is available at https://github.com/LFhase/GALA.

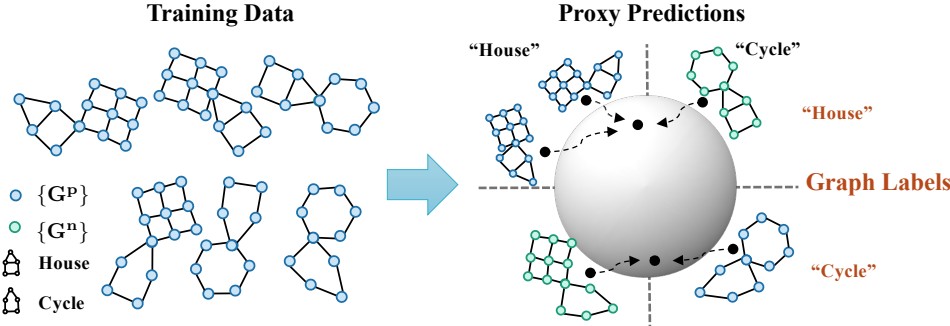

Figure 1: An illustration of GALA with the task of classifying graphs according to whether there exists a "House" or "Cycle" motif. Given the training data where the "House" subgraph often co-occurs with a "Grid" and the "Cycle" subgraph often co-occurs with a "Hexagon". An ERM trained environment assistant model will fit the spurious subgraph and therefore yield proxy predictions "House" or "Cycle" for any graphs containing a "Grid" (left half) or "Hexagon" (right half), respectively. GALA first separates the samples according to the correctness of the proxy predictions into the sets of positive graphs $\{G^p\}$ (correct, in blue) and negative graphs $\{G^n\}$ (incorrect, in green). Then, GALA extracts the maximally invariant subgraph among $\{G^p\}$ and $\{G^n\}$, i.e., pulling graphs with the same graph label but from $\{G^p\}$ and $\{G^n\}$ closer in the latent space, hence identifies the invariant subgraph.

distributions from different environments [8, 72]. Thus, the predictions made merely based on the invariant subgraphs can be generalized to OOD graphs that come from a new environment [60].

As the environment labels or partitions on graphs are often expensive to obtain [8], augmenting the environment information, such as generating new environments [45, 72, 73] and inferring the environment labels [41, 77], has become the *de facto* approach for invariant graph learning. However, little attention has been paid to verifying the *fidelity* (or *faithfulness*[2]) of the augmented environment information. For example, if the generated environments or inferred environment labels induce a higher bias or noise, it would make the learning of graph invariance even harder. Although it looks appealing to *learn both* the environment information and the graph invariance, the existing approaches could easily run into the "no free lunch" dilemma [71]. In fact, Lin et al. [43] found that there exist negative cases in the Euclidean regime where it is impossible to identify the invariant features without environment partitions. When it comes to the graph regime where the OOD generalization is fundamentally more difficult [8] than the Euclidean regime, it raises a challenging research question:

*When and how could one learn graph invariance without the environment labels?*

In this work, we present a theoretical investigation of the problem and seek a set of *minimal assumptions* on the underlying environments for feasible invariant graph learning. Based on a family of simple graph examples (Def. 3.1), we show that existing environment generation approaches can fail to generate faithful environments, when the underlying environments are not sufficient to uncover all the variations of the spurious subgraphs (Prop. 3.2). On the contrary, incorporating the generated environments can even lead to a worse OOD performance. The failure of faithful environment generation implies the necessity of *variation sufficiency* (Assumption 3.3). Moreover, even with sufficient environments, inferring faithful environment labels remains impossible. Since invariant and spurious subgraphs can have an arbitrary degree of correlation with labels, there exist multiple sets of training environments that have the same joint distribution of $P(G, Y)$ but different invariant subgraphs. Any invariant graph learning algorithms will inevitably fail to identify the invariant subgraph in at least one set of training environments (Prop. 3.4). Therefore, we need to additionally ensure the *variation consistency* (Assumption. 3.5), that is, the invariant and spurious subgraphs should have a consistent relationship in the correlation strengths with the labels.

To resolve the OOD generalization challenge under the established assumptions, we propose a new framework **G**raph inv**A**riant **L**earning **A**ssistant (GALA). GALA incorporates an additional assistant model that needs to be prone to distribution shifts, to generate proxy predictions of the training samples. Different from previous environment inferring approaches [41, 77], GALA does not require explicit environment labels but merely proxy predictions to differentiate the variations in the spurious

---

[2]The *fidelity* or *faithfulness* refers to whether the augmented environment information can actually improve the OOD generalization on graphs.

subgraphs. As shown in Fig. 1, we first fit an environment assistant model to the training distribution and then divide the training graphs into a positive set $\{G^p\}$ and a negative $\{G^n\}$, according to whether the proxy predictions are correct or not, respectively. As spurious correlations tend to vary more easily than invariant correlations, the variations in spurious subgraphs are further differentiated and increased between $\{G^p\}$ and $\{G^n\}$. Then, only the invariant subgraph holds an invariant correlation with the label among $\{G^p\}$ and $\{G^n\}$, and hence can be identified by extracting the subgraphs that maximize the intra-class subgraph mutual information among $\{G^p\}$ and $\{G^n\}$ (Theorem 4.1).

We conduct extensive experiments to validate the effectiveness of GALA using 12 datasets with various graph distribution shifts. Notably, GALA brings improvements up to $30\%$ in multiple graph datasets.

Our contributions can be summarized as follows:

- We identify failure cases of existing invariant graph learning approaches and establish the minimal assumptions for feasible invariant graph learning;
- We develop a novel framework GALA with provable identifiability of the invariant subgraph for OOD generalization on graphs under the assumptions;
- We conduct extensive experiments to verify both our theoretical results and the superiority of GALA;

Notably, both our theory and solution differ from Lin et al. [43] fundamentally, as we do not rely on the auxiliary information and are compatible with the existing interpretable and generalizable GNN architecture for OOD generalization on graphs. Meanwhile, we provide a new theoretical framework that resolves the counterexample in Lin et al. [43] while enjoying provable identifiability.

## 2  Background and Preliminaries

We begin by introducing the key concepts and backgrounds of invariant graph learning, and leave more details in Appendix C. The notations used in the paper are given in Appendix A.

**OOD generalization on graphs.** This work focuses on graph classification, while the results generalize to node classification as well using the same setting as in Wu et al. [72]. Specifically, we are given a set of graph datasets $\mathcal{D} = \{\mathcal{D}_e\}_{e \in \mathcal{E}_{\text{all}}}$ collected from multiple environments $\mathcal{E}_{\text{all}}$. Samples $(G_i^e, Y_i^e) \in \mathcal{D}^e$ from the environment $e$ are drawn independently from an identical distribution $\mathbb{P}^e$. The goal of OOD generalization on graphs is to find a GNN $f$ that minimizes the maximal loss among all environments, i.e., to minimize $\max_{e \in \mathcal{E}_{\text{all}}} R^e$, where $R^e$ is the risk of $f$ under environment $e$. We consider the same graph generation process proposed by Chen et al. [8] which is inspired by real-world drug discovery task [57] and covers a broad case of graph distribution shifts. As shown in Fig. 2, the generation of the observed graphs $G$ and labels $Y$ are controlled by a latent causal variable $C$ and a spurious variable $S$. $C$ and $S$ control $Y$ and $G$ by controlling the generation of the underlying invariant subgraph $G_c$ and spurious subgraph $G_s$, respectively. Since $S$ can be affected by the environment $E$, the correlation between $Y$ and $G_s$ can change arbitrarily when the environment changes. Besides, the interaction among $C$, $S$ and $Y$ at the latent space can be further categorized into *Full Informative Invariant Features* (*FIIF*) when $Y \perp\!\!\!\perp S|C$, and *Partially Informative Invariant Features* (*PIIF*) when $Y \not\perp\!\!\!\perp S|C$.

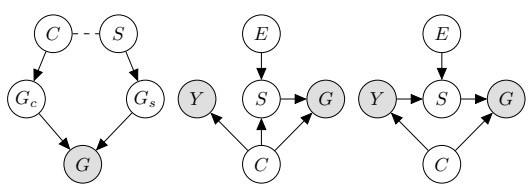

(a) $\mathcal{G}$-Gen. SCM  (b) FIIF SCM  (c) PIIF SCM

Figure 2: SCMs on graph distribution shifts [8].

To tackle the OOD generalization challenge on graphs from Fig. 2, the existing invariant graph learning approaches are generically designed to identify the underlying invariant subgraph $G_c$ to predict the label $Y$ [8, 72]. Specifically, the goal of OOD generalization on graphs is to learn an *invariant GNN* $f := f_c \circ g$, which is composed of: a) a featurizer $g : \mathcal{G} \to \mathcal{G}_c$ that estimates the invariant subgraph $\widehat{G}_c$; b) a classifier $f_c : \mathcal{G}_c \to \mathcal{Y}$ that predicts the label $Y$ based on the extracted $\widehat{G}_c$, where $\mathcal{G}_c$ refers to the space of subgraphs of $\mathcal{G}$. The learning objectives of $f_c$ and $g$ are formulated as

$$\max_{f_c, g} I(\widehat{G}_c; Y), \text{ s.t. } \widehat{G}_c \perp\!\!\!\perp E, \ \widehat{G}_c = g(G). \tag{1}$$

Since $E$ is not observed, many strategies are proposed to impose the independence of $\widehat{G}_c$ and $E$. A prevalent approach is to augment the environment information. Based on the estimated invariant

subgraphs $\widehat{G}_c$ and spurious subgraphs $\widehat{G}_s$, Liu et al. [45], Wu et al. [72, 73] propose to generate new environments, while Li et al. [41], Yang et al. [77] propose to infer the underlying environment labels. However, we show that they all fail to augment faithful environment information in Sec. 3.

Besides, Miao et al. [55, 56], Yu et al. [81, 82, 83] adopt graph information bottleneck to tackle FIIF graph shifts, but they cannot generalize to PIIF shifts, while Our work focuses on PIIF shifts as it is more challenging when without environment labels [43]. Fan et al. [17] generalize [40] to tackle severe graph biases, i.e., when $H(S|Y) < H(C|Y)$. Chen et al. [8] propose a contrastive framework to tackle both FIIF and PIFF graph shifts, but is limited to $H(S|Y) > H(C|Y)$. In practice, as it is usually unknown which correlation is stronger, we need a unified solution to tackle both cases.

**Invariant learning without environment labels.** In the Euclidean regime, there are plentiful studies in invariant learning without environment labels. Creager et al. [12] propose a minmax formulation to infer the environment labels. Liu et al. [46] propose a self-boosting framework based on the estimated invariant and variant features. Deng et al. [14], Liu et al. [44], Pezeshki et al. [61], Zhang et al. [85] propose to infer labels based on the failures of an ERM model. However, Lin et al. [43] find failure cases of the aforementioned approaches that it is impossible to identify the invariant features without given environment labels in Euclidean data, and propose a solution that leverages auxiliary environment information for invariant learning. As the OOD generalization on graphs poses more challenges [8], whether it is feasible to learn invariant graph representations without any auxiliary environment information remains elusive.

## 3 Pitfalls of Environment Augmentation

Given only the mixed training data without environment partitions, is it possible to learn to generate faithful environments or infer the underlying environment labels that facilitate OOD generalization on graphs? In the discussion below, we adopt the two-piece graphs to instantiate the problem, which is the simplistic version of the PIIF distribution shifts in Fig. 2(c), motivated by Kamath et al. [30].

**Definition 3.1** (Two-piece graphs). *Each environment $e$ is defined with two parameters, $\alpha_e, \beta_e \in [0, 1]$, and the dataset $(G^e, Y^e) \in \mathcal{D}_e$ is generated as follows:*

(a) *Sample $Y^e \in \{-1, 1\}$ uniformly;*

(b) *Generate $G_c$ and $G_s$ via : $G_c := f_{gen}^{G_c}(Y^e \cdot Rad(\alpha_e))$, $G_s := f_{gen}^{G_s}(Y^e \cdot Rad(\beta_e))$, where $f_{gen}^{G_c}, f_{gen}^{G_s}$ map the input $\{-1, 1\}$ to a corresponding graph selected from a given set, and $Rad(\alpha)$ is a random variable taking value $-1$ with probability $\alpha$ and $+1$ with $1 - \alpha$;*

(c) *Synthesize $G^e$ by randomly assembling $G_c$ and $G_s$: $G^e := f_{gen}^G(G_c, G_s)$.*

We denote an environment $e$ with $(\alpha, \beta_e)$ for simplicity. Different environments will have a different $\beta_e$, thus $P(Y|G_s)$ will change across different environments, while $P(Y|G_c)$ remains invariant.

### 3.1 Pitfalls of environment generation

We begin by discussing the cases where there are few environments, and generating new environments is necessary [45, 72, 73]. Environment generation aims to provide some additional "virtual" environments $\mathcal{E}_v$ such that the invariant subgraph can be identified via applying an OOD risk to the joint dataset with the augmented data $\mathcal{D}_{tr}^v = \{\mathcal{D}_e | e \in \mathcal{E}_{tr} \cup \mathcal{E}_v\}$.

The generation of "virtual" environments is primarily based on the intermediate estimation of the invariant and spurious subgraphs, denoted as $\widehat{G}_c$ and $\widehat{G}_s$, respectively. Liu et al. [45], Wu et al. [73] propose DIR and GREA to construct new graphs by assembling $\widehat{G}_c$ and $\widehat{G}_s$ from different graphs. Specifically, given $n$ samples $\{G^i, Y^i\}_{i=1}^n$,[3] the new graph samples in $\mathcal{E}_v$ is generated as follows:

$$G^{i,j} = f_{gen}^G(\widehat{G}_c^i, \widehat{G}_s^j), \ \forall i, j \in \{1...n\}, \ Y^{i,j} = Y^i,$$

which generates a new environment $\mathcal{E}_v$ with $n^2$ samples. Although both DIR and GREA gain some empirical success, the faithfulness of $\mathcal{E}_v$ remains questionable, as the generation is merely based on

---

[3]We slightly abuse the superscript and subscript when denoting the $i$th sample to avoid confusion of double superscripts or subscripts.

*inaccurate* estimations of the invariant and spurious subgraphs. Specifically, when $\widehat{G}_c$ contains parts of $G_s$, assigning the same labels to the generated graph is more likely to *strengthen* the spurious correlation between $G_s$ and $Y$. For example, when the model yields a reversed estimation, i.e., $\widehat{G}_c = G_s$ and $\widehat{G}_s = G_c$, the generated environment will destroy the invariant correlations.

**Proposition 3.2.** *Consider the two-piece graph dataset $\mathcal{E}_{tr} = \{(\alpha, \beta_1), (\alpha, \beta_2)\}$ with $\alpha \geq \beta_1, \beta_2$ (e.g., $\mathcal{E}_{tr} = \{(0.25, 0.1), (0.25, 0.2)\}$), and its corresponding mixed environment $\mathcal{E}_{tr}^{mix} = \{(\alpha, (\beta_1 + \beta_2)/2)\}$ (e.g., $\mathcal{E}_{tr}^{mix} = \{(0.25, 0.15)\}$). When $\widehat{G}_c = G_s$ and $\widehat{G}_s = G_c$, it holds that the augmented environment $\mathcal{E}_v$ is also a two-piece graph dataset with*

$$\mathcal{E}_v = \{(0.5, (\beta_1 + \beta_2)/2)\} \ (e.g., \ \mathcal{E}_v = \{(0.5, 0.15)\}).$$

The proof is given in Appendix E.1. This also extends to the adversarial augmentation [72, 83], which will destroy the actual $\widehat{G}_c$. As both `DIR` and `GREA` adopt the same environment generation procedure, we verify the failures of environment generation with `GREA` in Table 2 of Sec. 5, where `GREA` can perform comparably with ERM. In fact, when the underlying environments are insufficient to differentiate the variations of the spurious features, it is fundamentally impossible to identify the underlying invariant graph from the spurious subgraph. More formally, if $\exists G_s$, such that $P^{e_1}(Y|G_s) = P^{e_2}(Y|G_s)$ for any $e_1, e_2 \in \mathcal{E}_{tr}$, where $P^e(Y|G_s)$ is the conditional distribution $P(Y|G_s)$ under environment $e \in \mathcal{E}_{all}$, it is impossible for any graph learning algorithm to identify $G_c$. We provide a formal discussion in Appendix E.2. The failure implies a fundamental requirement that $\mathcal{E}_{tr}$ should uncover all the potential variations in the spurious subgraph.

**Assumption 3.3.** *(Variation sufficiency) For graphs generated following Fig. 2, for any $G_s$, $\exists e_1, e_2 \in \mathcal{E}_{tr}$, such that $P^{e_1}(Y|G_s) \neq P^{e_2}(Y|G_s)$, and $P^{e_1}(Y|G_c) = P^{e_2}(Y|G_c)$.*

Assumption 3.3 aligns with the definition of invariance [8, 30] that the invariant subgraph $G_c$ is expected to satisfy $P^{e_1}(Y|G_c) = P^{e_2}(Y|G_c)$ for $e_1, e_2 \in \mathcal{E}_{all}$. If there exists $G_s$ satisfying the invariance condition as well, then it is impossible to tell $G_c$ from $G_s$ even with environment labels.

## 3.2 Pitfalls of environment inferring

Although environment sufficiency (Assumption 3.3) relieves the need for generating new environments, is it possible to infer the underlying environment labels via approaches such as `MoleOOD` [77] and `GIL` [41], to facilitate invariant graph learning? Unfortunately, we find a negative answer.

Considering the two-piece graph examples $\mathcal{E}_{tr} = \{(0.2, 0.1), (0.2, 0.3)\}$, when given the underlying environment labels, it is easy to identify the invariant subgraphs from spurious subgraphs. However, when the environment labels are not available, we have the mixed data as $\mathcal{E}_{tr} = \{(0.2, 0.2)\}$, where $P(Y|G_c) = P(Y|G_s)$. The identifiability of $G_s$ is *ill-posed*, as it does not affect the $\mathcal{E}_{tr}$ even if we swap $G_c$ and $G_s$. More formally, considering the environment mixed from two two-piece graph environments $\{(\alpha, \beta_1)\}$ and $\{(\alpha, \beta_2)\}$, then we have $\mathcal{E}_{tr} = \{(\alpha, (\beta_1 + \beta_2)/2)\}$. For each $\mathcal{E}_{tr}$, we can also find a corresponding $\mathcal{E}_{tr}' = \{((\beta_1' + \beta_1')/2, \alpha')\}$ with $\{(\beta_1', \alpha')\}$ and $\{(\beta_2', \alpha')\}$. Then, let

$$\alpha = (\beta_1' + \beta_1')/2 = \alpha' = (\beta_1 + \beta_2)/2. \tag{2}$$

We now obtain $\mathcal{E}_{tr}$ and $\mathcal{E}_{tr}'$ which share the same joint distribution $P(Y, G)$ while the underlying $G_c$ is completely different. More generally, we have the following proposition.

**Proposition 3.4.** *There exist 2 two-piece graph training environments $\mathcal{E}_{tr}$ and $\mathcal{E}_{tr}'$ that share the same joint distribution $P(Y, G)$. Any learning algorithm will fail in either $\mathcal{E}_{tr}$ or $\mathcal{E}_{tr}'$.*

The proof is given in Appendix E.3. The experiments in Sec. 5 validate that both `MoleOOD` and `GIL` fail to infer faithful environment labels and even underperform ERM. It implies that whenever it allows the existence of an identical training distribution by mixing the environments, invariant graph learning is impossible. Therefore, we need an additional assumption that excludes the unidentifiable case. We propose to constrain the relationship between $\alpha$ (i.e., $H(Y|G_c)$ ) and $\beta_e$ (i.e., $H(Y|G_s)$).

**Assumption 3.5.** *(Variation consistency) For all environments in $\mathcal{E}_{tr}$, $H(C|Y) \neq H(S|Y)$.*

Intuitively, Assumption 3.5 imposes the consistency requirement on the correlation strengths between invariant and spurious subgraphs with labels. For two-piece graphs with consistent variations, mixing up the environments will yield a new environment with the same variation strength relationships.

Thus, Assumption 3.5 gets rid of the previous unidentifiable cases. Moreover, Assumption 3.5 also aligns with many realistic cases. For example, the relation of a specific functional group (e.g., -OH) with a molecule can hardly be reversed to that held upon the scaffold of the molecule, due to the data collection process. Therefore, Assumption 3.5 also resolves the counterexample proposed by Lin et al. [43]. Different from our work, Lin et al. [43] propose to incorporate additional auxiliary information that satisfies certain requirements to mitigate the unidentifiable case. However, such auxiliary information is often unavailable and expensive to obtain on graphs. More importantly, the requirements are also unverifiable without more assumptions, which motivates us to consider the relaxed case implied by Assumption 3.5.

### 3.3 Challenges of environment augmentation

To summarize, the two assumptions constitute the minimal assumptions for feasible invariant graph learning. Failing to satisfy either one of them while lacking additional inductive biases will result in the "no free lunch" dilemma [71] and suffer from the unidentifiability issue.

**Corollary 3.6.** *(No Free Graph OOD Lunch) Without Assumption 3.3 or Assumption 3.5, there does not exist a learning algorithm that captures the invariance of the two-piece graph environments.*

Corollary 3.6 is a natural conclusion from the previous discussion. The proof is straightforward and given in Appendix E.4. Assumption 3.3 and Assumption 3.5 establish the minimal premises for identifying the underlying invariant subgraphs. However, it also raises new challenges, as shown in Table. 1. Chen et al. [8]

Table 1: Remaining challenges of invariant graph learning: no existing works can handle both cases.

|  | $H(S|Y) < H(C|Y)$ | $H(S|Y) > H(C|Y)$ |
|---|---|---|
| DisC | ✓ | ✗ |
| CIGA | ✗ | ✓ |
| GALA (Ours) | ✓ | ✓ |

propose CIGA to maximize the intra-class mutual information of the estimated invariant subgraphs to tackle the case when $H(C|Y) < H(S|Y)$. While for the case when $H(S|Y) < H(C|Y)$, Fan et al. [17] propose DisC that adopts GCE loss [40] to extract the spurious subgraph with a larger learning step size such that the left subgraph is invariant. However, both of them can fail when there is no prior knowledge about the relations between $H(C|Y)$ and $H(S|Y)$. We verify the failures of DisC and CIGA in Table. 2. The failure thus raises a challenging question:

*Given the established minimal assumptions, is there a unified framework that tackles both cases when $H(C|Y) < H(S|Y)$ and $H(C|Y) > H(S|Y)$?*

## 4 Learning Invariant Graph Representations with Environment Assistant

We give an affirmative answer by proposing a new framework, GALA: **G**raph inv**A**riant **L**earning **A**ssistant, which adopts an assistant model to provide proxy information about the environments.

### 4.1 Learning with An Environment Assistant

Intuitively, a straightforward approach to tackle the aforementioned challenge is to extend the framework of either DisC [17] or CIGA [8] to resolve the other case. As DisC always destroys the first learned features and tends to be more difficult to extend (which is empirically verified in Sec. 5), we are motivated to extend the framework of CIGA to resolve the case when $H(S|Y) < H(C|Y)$.

**Understanding the success and failure of CIGA.** The principle of CIGA lies in maximizing the intra-class mutual information of the estimated invariant subgraphs, i.e.,

$$\max_{f_c,g} I(\widehat{G}_c; Y), \text{ s.t. } \widehat{G}_c \in \arg\max_{\widehat{G}_c=g(G),|\widehat{G}_c|\leq s_c} I(\widehat{G}_c; \widehat{G}_c^s|Y), \tag{3}$$

where $\widehat{G}_c^s = g(G^s)$ and $G^s \sim \mathbb{P}(G|Y)$, i.e., $\widehat{G}$ is sampled from training graphs that share the same label $Y$ as $\widehat{G}$. The key reason for the success of Eq. 3 is that, given the data generation process as in Fig. 2 and the same $C$, the underlying invariant subgraph $G_c$ maximizes the mutual information of subgraphs from any two environments, i.e., $\forall e_1, e_2 \in \mathcal{E}_{\text{all}}$,

$$G_c^{e_1} \in \arg\max_{\widehat{G}_c^{e_1}} I(\widehat{G}_c^{e_1}; \widehat{G}_c^{e_2}|C), \tag{4}$$

where $\widehat{G}_c^{e_1}$ and $\widehat{G}_c^{e_2}$ are the estimated invariant subgraphs corresponding to the same latent causal variable $C = c$ under the environments $e_1, e_2$, respectively. Since $C$ is not observable, `CIGA` adopts $Y$ as a proxy for $C$, as when $H(S|Y) > H(C|Y)$, $G_c$ maximizes $I(\widehat{G}_c^{e_1}; \widehat{G}_c^{e_2}|Y)$ and thus $I(\widehat{G}_c; \widehat{G}_c^s|Y)$. However, when $H(S|Y) < H(C|Y)$, the proxy no longer holds. Given the absence of $E$, simply maximizing intra-class mutual information favors the spurious subgraph $G_s$ instead, i.e.,

$$G_s \in \arg\max_{\widehat{G}_c} I(\widehat{G}_c; \widehat{G}_c^s|Y). \tag{5}$$

**Invalidating spuriousness dominance.** To mitigate the issue, we are motivated to find a new proxy that samples $\widehat{G}_c$ for Eq. 5, while preserving only the $G_c$ as the solution under both cases.

To begin with, we consider the case of $H(S|Y) < H(C|Y)$. Although the correlation between $G_s$ and $Y$ dominates the intra-class mutual information, Assumption 3.3 implies that there exists a subset of training data where $P(Y|G_s)$ varies, while $P(Y|G_c)$ remains invariant. Therefore, the dominance of spurious correlations no longer holds for samples from the subset. Incorporating samples from the subset into Eq. 3 as $\widehat{G}_c^s$ invalidates the dominance of $G_s$. Denote the subset as $\{\widehat{G}_c^n\}$, then

$$G_c \in \arg\max_{\widehat{G}_c^p} I(\widehat{G}_c^p; \widehat{G}_c^n|Y), \tag{6}$$

where $\widehat{G}_c^p \in \{\widehat{G}_c^p\}$ is sampled from the subset $\{\widehat{G}_c^p\}$ dominated by spurious correlations, while $\widehat{G}_c^n \in \{\widehat{G}_c^n\}$ is sampled from the subset $\{\widehat{G}_c^n\}$ where spurious correlation no long dominates, or is dominated by invariant correlations. We prove the effectiveness of Eq. 6 in Theorem 4.1.

**Environment assistant model $A$.** To find the desired subsets $\{\widehat{G}_c^p\}$ and $\{\widehat{G}_c^n\}$, inspired by the success in tackling spuriousness-dominated OOD generalization via learning from a biased predictors [40, 44, 58, 85], we propose to incorporate an assistant model $A$ that is prone to spurious correlations. Simply training $A$ with ERM using the spuriousness-dominated data enables $A$ to learn spurious correlations, and hence identifies the subsets where the spurious correlations hold or shift, according to whether the predictions of $A$ are correct or not, respectively. Let $A = \arg\max_{\widehat{A}} I(\widehat{A}(G); Y)$, we have

$$\{\widehat{G}_c^p\} = \{g(G_i^p)|A(G_i^p) = Y_i\}, \quad \{\widehat{G}_c^n\} = \{g(G_i^n)|A(G_i^n) \neq Y_i\}. \tag{7}$$

**Reducing to invariance dominance case.** After showing that Eq. 6 resolves the spuriousness dominance case, we still need to show that Eq. 6 preserves $G_c$ as the only solution when $H(S|Y) > H(C|Y)$. Considering training $A$ with ERM using the invariance-dominated data, $A$ will learn both invariant correlations and spurious correlations [10, 17]. Therefore, $\{\widehat{G}_c^n\}$ switches to the subset that is dominated by spurious correlations, while $\{\widehat{G}_c^p\}$ switches to the subset dominated by invariant correlations. Then, Eq. 6 establishes a lower bound for the intra-class mutual information, i.e.,

$$I(\widehat{G}_c^p; \widehat{G}_c^n|Y) \leq I(\widehat{G}_c; \widehat{G}_c^s|Y), \tag{8}$$

where $\widehat{G}_c^p \in \{\widehat{G}_c^p\}, \widehat{G}_c^n \in \{\widehat{G}_c^n\}$, and $\widehat{G}_c, \widehat{G}_c^s$ are the same as in Eq. 3. The inequality in Eq. 8 holds as any subgraph maximizes the left hand side can also be incorporated in right hand side, while the sampling space of $\widehat{G}_c$ and $\widehat{G}_c^s$ in the right hand side (i.e., both $\widehat{G}_c$ and $\widehat{G}_c^s$ are sampled from the whole train set) is larger than that of the left hand side. The equality is achieved by taking the ground truth $G_c$ as the solution for the featurizer $g$. We verify the correctness of Eq. 6 and Eq. 8 in Fig. 3(a).

---

**Algorithm 1 GALA**: **G**raph inv**A**riant **L**earning **A**ssistant

1: **Input:** Training data $\mathcal{D}_{tr}$; environment assistant $A$; featurizer GNN $g$; classifier GNN $f_c$; length of maximum training epochs $e$; batch size $b$;
2: Initialize environment assistant $A$;
3: **for** $p \in [1, \ldots, e]$ **do**
4:   Sample a batch of data $\{G_i, Y_i\}_{i=1}^b$ from $\mathcal{D}_{tr}$;
5:   Obtain Environment Assistant predictions $\{\hat{y}_i^e\}_{i=1}^b$;
6:   **for** each sample $G_i, y_i \in \{G_i, Y_i\}_{i=1}^b$ **do**
7:     Find *positive* graphs with same $y_i$ and different $\hat{y}_i^e$;
8:     Find *negative* graphs with different $y_i$ but same environment assistant prediction $\hat{y}_i^e$;
9:     Calculate `GALA` risk via Eq. 9;
10:    Update $f_c, g$ via gradients from `GALA` risk;
11:   **end for**
12: **end for**
13: **return** final model $f_c \circ g$;

---

### 4.2 Practical implementations.

The detailed algorithm description of `GALA` is shown as in Algorithm 1. In practice, the environment assistant can have multiple implementation choices so long as it is prone to distribution shifts. As discussed in Sec. 4.1, ERM trained

model can serve as a reliable environment assistant, since ERM tends to learn the dominant features no matter whether the features are invariant or spurious. For example, when $H(S|Y) < H(C|Y)$, ERM will first learn to use spurious subgraphs $G_s$ to make predictions. Therefore, we can obtain $\{G^p\}$ by finding samples where ERM correctly predicts the labels, and $\{G^n\}$ for samples where ERM predicts incorrect labels. In addition to label predictions, the clustering predictions of the hidden representations yielded by environment assistant models can also be used for sampling $\{G^p\}$ and $\{G^n\}$ [85]. Besides, we can also incorporate models that are easier to overfit to the first dominant features to better differentiate $\{G^p\}$ from $\{G^n\}$. When the number of positive or negative samples is imbalanced, we can upsample the minor group to avoid trivial solutions. In addition, the final GALA objective is given in Eq. 9 and implemented as in Eq. 18. We provide more discussions about the implementation options in Appendix F.

### 4.3 Theoretical analysis

In the following theorem, we show that the GALA objective derived in Sec. 4.1 can identify the underlying invariant subgraph and yields an invariant GNN defined in Sec. 2.

**Theorem 4.1.** *Given i) the same data generation process as in Fig. 2; ii) $\mathcal{D}_{\mathrm{tr}}$ that satisfies variation sufficiency (Assumption 3.3) and variation consistency (Assumption 3.5); iii) $\{G^p\}$ and $\{G^n\}$ are distinct subsets of $\mathcal{D}_{\mathrm{tr}}$ such that $I(G_s^p; G_s^n|Y) = 0$, $\forall G_s^p = \arg\max_{\widehat{G}_s^p} I(\widehat{G}_s^p; Y)$ under $\{G^p\}$, and $\forall G_s^n = \arg\max_{\widehat{G}_s^n} I(\widehat{G}_s^n; Y)$ under $\{G^n\}$; suppose $|G_c| = s_c$, $\forall G_c$, resolving the following GALA objective elicits an invariant GNN defined via Eq. 1,*

$$\max_{f_c, g} I(\widehat{G}_c; Y), \ s.t. \ g \in \arg\max_{\hat{g}, |\widehat{G}_c^p| \le s_c} I(\widehat{G}_c^p; \widehat{G}_c^n | Y), \tag{9}$$

*where $\widehat{G}_c^p \in \{\widehat{G}_c^p = g(G^p)\}$ and $\widehat{G}_c^n \in \{\widehat{G}_c^n = g(G^n)\}$ are the estimated invariant subgraphs via $g$ from $\{G^p\}$ and $\{G^n\}$, respectively.*

The proof is given in Appendix E.5. Essentially, assumption iii) in Theorem 4.1 is an implication of the variation sufficiency (Assumption 3.3). When given the distinct subsets $\{G^p\}$ and $\{G^n\}$ with different relations of $H(C|Y)$ and $H(S|Y)$, since $H(C|Y)$ remains invariant across different subsets, the variation happens mostly to the spurious correlations between $S$ and $Y$. By differentiating spurious correlations into distinct subsets, maximizing the intra-class mutual information helps identify the true invariance. The fundamental rationale for why GALA resolves two seemingly conversed cases essentially relies on the commutative law of mutual information.

## 5 Experiments

We evaluated GALA with both synthetic and realistic graph distribution shifts. Specifically, we are interested in the following two questions: (a) Can GALA improve over the state-of-the-art invariant graph learning methods when the spurious subgraph has a stronger correlation with the labels? (b) Will GALA affect the performance when the invariant correlations are stronger?

### 5.1 Datasets and experiment setup

We prepare both synthetic and realistic graph datasets containing various distribution shifts to evaluate GALA. We will briefly introduce each dataset and leave more details in Appendix G.1.

**Two-piece graph datasets.** We adopt BA-2motifs [50] to implement 4 variants of 3-class two-piece graph (Def. 3.1) datasets. The datasets contain different relationships of $H(C|Y)$ and $H(S|Y)$ by controlling the $\alpha$ and $\beta$ in the mixed environment, respectively. We consider 4 cases of $\alpha - \beta$, ranging from $\{+0.2, +0.1, -0.1, -0.2\}$, to verify our discussion in Sec. 4.3.

**Realistic datasets.** We also adopt datasets containing various realistic graph distribution shifts to comprehensively evaluate the OOD performance of GALA. We adopt 6 datasets from DrugOOD benchmark [28], which focuses on the challenging real-world task of AI-aided drug affinity prediction. The DrugOOD datasets include splits using Assay, Scaffold, and Size from the EC50 category (denoted as **EC50-\***) and the Ki category (denoted as **Ki-\***). We also adopt graphs converted from the ColoredMNIST dataset [3] using the algorithm from Knyazev et al. [36], which contains distribution

shifts in node attributes (denoted as **CMNIST-sp**). In addition, we adopt **Graph-SST2** [84], where we split graphs with a larger average degree in the training set while smaller in the test set.

**Experiment setup.** We adopt the state-of-the-art OOD methods from the Euclidean regime, including IRMv1 [3], VREx [38], EIIL [13] and IB-IRM [2], and from the graph regime, including GREA [45], GSAT [55], CAL [64], MoleOOD [77], GIL [41], DisC [17] and CIGA [8]. We exclude DIR [73] and GIB [81] as GREA and GSAT are their sophisticated variants. In addition to the ERM baseline that trained a vanilla GNN with ERM objective, in two-piece motif datasets, we also include XGNN to demonstrate the failures of previous approaches, which is an interpretable GNN trained with ERM. We also exclude CIGAv2 [8] as `GALA` focuses on improving the contrastive sampling via environment assistant for the objective in CIGAv1. All methods use the same GIN backbone [75], the same interpretable GNN architecture as in [55], and optimization protocol for fair comparisons. We tune the hyperparmeters following the common practice. Details are given in Appendix G.2.

## 5.2 Experimental results and analysis

**Proof-of-concept study.** The results in two-piece graph datasets are reported in Table 2. It can be found that the previous environment augmentation approaches fail either in datasets where the invariant correlations dominate or where the spurious correlations dominate, aligned with our discussions in Sec. 3. In particular, `GREA`, `CIGA` and `GIL` achieve high performance when the invariant correlation dominates, but suffer great performance decrease when the spurious correlations are stronger. Although `DisC` is expected to succeed when spurious correlations dominate, `DisC` fails to outperform others because of its excessive destruction of the learned information. `MoleOOD` also yields degraded performance, which

Table 2: OOD generalization performance under various invariant and spurious correlation degrees in the two-piece graph datasets. Each dataset is generated from a variation of two-piece graph model, denoted as $\{a, b\}$, where $a$ refers to the invariant correlation strength and $b$ refers to the spurious correlation strength. The blue shadowed entries are the results with the mean-1*std larger than the mean of the corresponding second best results.

| Datasets | $\{0.8, 0.6\}$ | $\{0.8, 0.7\}$ | $\{0.8, 0.9\}$ | $\{0.7, 0.9\}$ | Avg. |
|---|---|---|---|---|---|
| ERM | $77.33_{\pm0.47}$ | $75.65_{\pm1.62}$ | $51.37_{\pm1.20}$ | $42.73_{\pm3.82}$ | 61.77 |
| IRM | $78.32_{\pm0.70}$ | $75.13_{\pm0.77}$ | $50.76_{\pm2.56}$ | $41.32_{\pm2.50}$ | 61.38 |
| V-Rex | $77.69_{\pm0.38}$ | $74.96_{\pm1.40}$ | $49.47_{\pm3.36}$ | $41.65_{\pm2.78}$ | 60.94 |
| IB-IRM | $78.00_{\pm0.68}$ | $73.93_{\pm0.79}$ | $50.93_{\pm1.87}$ | $42.05_{\pm0.79}$ | 61.23 |
| EIIL | $76.98_{\pm1.24}$ | $74.25_{\pm1.74}$ | $51.45_{\pm4.92}$ | $39.71_{\pm2.64}$ | 60.60 |
| XGNN | $83.84_{\pm0.59}$ | $83.05_{\pm0.20}$ | $53.37_{\pm1.32}$ | $38.28_{\pm1.71}$ | 64.63 |
| GREA | $82.86_{\pm0.50}$ | $82.72_{\pm0.50}$ | $50.34_{\pm1.74}$ | $39.01_{\pm1.21}$ | 63.72 |
| GSAT | $80.54_{\pm0.88}$ | $78.11_{\pm1.23}$ | $48.63_{\pm2.18}$ | $36.62_{\pm0.87}$ | 63.32 |
| CAL | $76.98_{\pm6.03}$ | $62.95_{\pm8.58}$ | $51.57_{\pm6.33}$ | $46.23_{\pm3.93}$ | 59.43 |
| MoleOOD | $49.93_{\pm2.25}$ | $49.85_{\pm7.31}$ | $38.49_{\pm4.25}$ | $34.81_{\pm1.65}$ | 43.27 |
| GIL | $83.51_{\pm0.41}$ | $82.67_{\pm1.18}$ | $51.76_{\pm4.32}$ | $40.07_{\pm2.61}$ | 64.50 |
| DisC | $60.47_{\pm17.9}$ | $54.29_{\pm15.0}$ | $45.06_{\pm7.82}$ | $39.42_{\pm8.59}$ | 50.81 |
| CIGA | $84.03_{\pm0.53}$ | $83.21_{\pm0.30}$ | $57.87_{\pm3.38}$ | $43.62_{\pm3.20}$ | 67.18 |
| **GALA** | $\mathbf{84.27}_{\pm0.34}$ | $\mathbf{83.65}_{\pm0.44}$ | $\mathbf{76.42}_{\pm3.53}$ | $\mathbf{72.50}_{\pm1.06}$ | **79.21** |
| Oracle | $84.73_{\pm0.36}$ | $85.42_{\pm0.25}$ | $84.28_{\pm0.15}$ | $78.38_{\pm0.19}$ | |

could be caused by the failures to infer reliable environment labels. In contrast, `GALA` achieves consistently high performance under *both* cases and improves CIGA up to 30% under $\{0.7, 0.9\}$ and 13% in average, which validates our theoretical results in Sec. 4.3.

Table 3: OOD generalization performance under realistic graph distribution shifts. The blue shadowed entries are the results with the mean-1*std larger than the mean of the respective second best results.

| Datasets | EC50-Assay | EC50-Sca | EC50-Size | Ki-Assay | Ki-Sca | Ki-Size | CMNIST-sp | Graph-SST2 | Avg.(Rank)[†] |
|---|---|---|---|---|---|---|---|---|---|
| ERM | $76.42_{\pm1.59}$ | $64.56_{\pm1.25}$ | $61.61_{\pm1.52}$ | $74.61_{\pm2.28}$ | $69.38_{\pm1.65}$ | $76.63_{\pm1.34}$ | $21.56_{\pm5.38}$ | $81.54_{\pm1.13}$ | 65.79 (6.50) |
| IRM | $77.14_{\pm2.55}$ | $64.32_{\pm0.42}$ | $62.33_{\pm0.86}$ | $75.10_{\pm3.38}$ | $69.32_{\pm1.84}$ | $76.25_{\pm0.73}$ | $20.25_{\pm3.12}$ | $82.52_{\pm0.79}$ | 65.91 (6.13) |
| V-Rex | $75.57_{\pm2.17}$ | $64.73_{\pm0.53}$ | $62.80_{\pm0.89}$ | $74.16_{\pm1.46}$ | $71.40_{\pm2.77}$ | $76.68_{\pm1.35}$ | $30.71_{\pm11.8}$ | $81.11_{\pm1.37}$ | 67.15 (5.25) |
| IB-IRM | $64.70_{\pm2.50}$ | $62.62_{\pm2.05}$ | $58.28_{\pm0.99}$ | $71.98_{\pm3.26}$ | $69.55_{\pm1.66}$ | $70.71_{\pm1.95}$ | $23.58_{\pm7.96}$ | $81.56_{\pm0.82}$ | 62.87 (10.6) |
| EIIL | $64.20_{\pm5.40}$ | $62.88_{\pm2.75}$ | $59.58_{\pm0.96}$ | $74.24_{\pm2.48}$ | $69.63_{\pm1.46}$ | $76.56_{\pm1.37}$ | $23.55_{\pm7.68}$ | $82.46_{\pm1.48}$ | 64.14 (8.00) |
| XGNN | $72.99_{\pm2.56}$ | $63.62_{\pm1.35}$ | $62.55_{\pm0.81}$ | $72.40_{\pm3.05}$ | $72.01_{\pm1.34}$ | $73.15_{\pm2.83}$ | $20.96_{\pm8.00}$ | $82.55_{\pm0.65}$ | 65.03 (7.13) |
| GREA | $66.87_{\pm7.53}$ | $63.14_{\pm2.19}$ | $59.20_{\pm1.42}$ | $73.17_{\pm1.80}$ | $67.82_{\pm4.67}$ | $73.52_{\pm2.75}$ | $12.77_{\pm1.71}$ | $82.40_{\pm1.98}$ | 62.36 (10.1) |
| GSAT | $76.07_{\pm1.95}$ | $63.58_{\pm1.36}$ | $61.12_{\pm0.66}$ | $72.26_{\pm1.76}$ | $70.16_{\pm0.80}$ | $75.78_{\pm2.60}$ | $15.24_{\pm3.72}$ | $80.57_{\pm0.88}$ | 64.35 (8.63) |
| CAL | $75.10_{\pm2.71}$ | $64.79_{\pm1.58}$ | $63.38_{\pm0.88}$ | $75.22_{\pm1.73}$ | $71.08_{\pm4.83}$ | $72.93_{\pm1.71}$ | $23.68_{\pm4.68}$ | $82.38_{\pm1.01}$ | 66.07 (5.38) |
| DisC | $61.94_{\pm7.76}$ | $54.10_{\pm5.69}$ | $57.64_{\pm1.57}$ | $54.12_{\pm8.53}$ | $55.35_{\pm10.5}$ | $50.83_{\pm9.30}$ | $50.26_{\pm0.40}$ | $76.51_{\pm2.17}$ | 56.59 (12.4) |
| MoleOOD | $61.49_{\pm2.19}$ | $62.12_{\pm1.91}$ | $58.74_{\pm1.73}$ | $75.10_{\pm0.73}$ | $60.35_{\pm11.3}$ | $73.69_{\pm2.29}$ | $21.04_{\pm3.36}$ | $81.56_{\pm0.35}$ | 61.76 (10.0) |
| GIL | $70.56_{\pm4.46}$ | $61.59_{\pm3.16}$ | $60.46_{\pm1.91}$ | $75.25_{\pm1.14}$ | $70.07_{\pm4.31}$ | $75.76_{\pm2.23}$ | $12.55_{\pm1.26}$ | $83.31_{\pm0.50}$ | 63.69 (8.00) |
| CIGA | $75.03_{\pm2.47}$ | $65.41_{\pm1.16}$ | $64.10_{\pm1.08}$ | $73.95_{\pm2.50}$ | $71.87_{\pm3.32}$ | $74.46_{\pm2.32}$ | $15.83_{\pm2.56}$ | $82.93_{\pm0.63}$ | 65.45 (5.88) |
| **GALA** | $\mathbf{77.56}_{\pm2.88}$ | $\mathbf{66.28}_{\pm0.45}$ | $64.25_{\pm1.21}$ | $\mathbf{77.92}_{\pm2.48}$ | $73.17_{\pm0.88}$ | $\mathbf{77.40}_{\pm2.04}$ | $\mathbf{68.94}_{\pm0.56}$ | $83.60_{\pm0.66}$ | **73.64 (1.00)** |
| Oracle | $84.77_{\pm0.58}$ | $82.66_{\pm1.19}$ | $84.53_{\pm0.60}$ | $91.08_{\pm1.43}$ | $88.58_{\pm0.64}$ | $92.50_{\pm0.53}$ | $67.76_{\pm0.60}$ | $91.40_{\pm0.26}$ | |

[†]Averaged rank is also reported in the parentheses because of dataset heterogeneity. A lower rank is better.

**OOD generalization in realistic graphs.** The results in realistic datasets are reported in Table 3. Aligned with our previous discussion, existing environment augmentation approaches sometimes yield better performance than ERM, such as CAL in EC50-Size, `MoleOOD` in Ki-Assay, `GIL` in Graph-SST2, or `CIGA` in EC50-Size, however, inevitably fail to bring consistent improvements than ERM, due to the existence of failure cases. `DisC` is suspected to work only for graph distribution shifts on node features and bring impressive improvements in CMNIST-sp, but can destroy the learned information under more challenging settings. In contrast, `GALA` consistently outperform ERM by a non-trivial margin in all datasets. Notably, `GALA` achieves near oracle performance in CMNIST-sp and improves `CIGA` by $53\%$. The consistent improvements of `GALA` confirm the effectiveness of `GALA`.

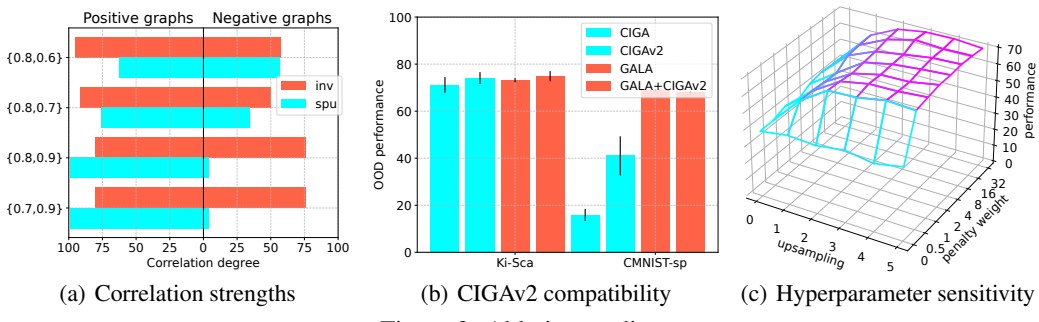

(a) Correlation strengths      (b) CIGAv2 compatibility      (c) Hyperparameter sensitivity

Figure 3: Ablation studies.

**Correlation strengths of** $\{G^p\}$ **and** $\{G^n\}$**.** We conduct experiments with the two-piece graph datasets evaluated in Table 2 to verify the correctness of Eq. 6 and Eq. 8. Eq. 6 and Eq. 8 imply that the underlying invariant subgraph will be the subgraph that maximizes the mutual information among subgraphs from $\{G^p\}$ and $\{G^n\}$, no matter whether the dominant correlation is spurious or not. We measure the invariant and spurious correlation strengths in terms of co-occur probability of the invariant and spurious subgraphs with the labels. The results are shown in Fig. 3(a). It can be found that, under both cases, the underlying invariant subgraph maintains the predictivity with the label in an invariant manner. Hence, maximizing the intra-class subgraph mutual information between $\{G^p\}$ and $\{G^n\}$ in `GALA` succeeds in identifying the underlying invariant subgraph.

**CIGAv2 compatibility.** Although `GALA` focuses on the contrastive term in CIGA, both `GALA` and CIGA are compatible with the additional CIGAv2 term that facilitates constraining the graph sizes. To verify, we compare the OOD performances of CIGA, CIGAv2, `GALA`, and `GALA` +CIGAv2 using two challenging datasets, Ki-Scaffold and CMNIST-sp. The results are given in Fig. 3(b). It can be found that, despite incorporating the additional CIGAv2 constraint, CIGA can not outperform `GALA`, while `GALA` can bring more improvements with the additional CIGAv2 constraint. In CMNIST-sp, since `GALA` already achieve the upper bound, incorporating CIGAv2 can only achieve a similar result.

**Hyperparameter sensitivity.** We also test the hyperparameter sensitivity of `GALA` to the contrastive penalty weights as well as the upsampling times that are introduced to mitigate the imbalance of positive and negative graphs. We conduct the experiments with two-piece graph dataset $\{0.7, 0.9\}$. As shown in Fig. 3(c), it can be found that `GALA` is generically robust to different hyperparameter choices. In addition, when the penalty weight or the upsampling times turn to $0$, the performance will decrease a lot, which serves as strong evidence for the effectiveness of `GALA`.

**Computational analysis.** We also conduct computational analysis of `GALA` and other methods, and defer the results to Table. 6 in Appendix G.4, due to space constraints. The results show that `GALA` costs only a competitive training time as environment generation based methods, while achieving much better OOD generalization performance.

## 6 Conclusions

We conducted a retrospective study on the faithfulness of the augmented environment information for OOD generalization on graphs. By showing hardness cases and impossibility results of the existing approaches, we developed a set of minimal assumptions for feasible invariant graph learning. Built upon the assumptions, we proposed `GALA` to learn the invariant graph representations guided by an environment assistant model. Extensive experiments with 12 datasets verified the superiority of `GALA`.

## Acknowledgements

We thank the reviewers for their valuable comments. This work was supported by CUHK direct grant 4055146. BH was supported by the NSFC Young Scientists Fund No. 62006202, NSFC General Program No. 62376235, Guangdong Basic and Applied Basic Research Foundation No. 2022A1515011652, HKBU Faculty Niche Research Areas No. RC-FNRA-IG/22-23/SCI/04, and Tencent AI Lab Rhino-Bird Gift Fund.

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

# Appendix of GALA

## Contents

# A   Notations

Typically, for graphs that appeared in the discussion, we will use the superscript to denote the sampling process (e.g., $G^p$ is the positive graph), and the subscript to denote the specific invariant (i.e., $G_c$) or spurious subgraph (i.e., $G_s$). Graph symbols with $\widehat{G}$ are the predicted graphs of a model (i.e., the estimated invariant subgraph $\widehat{G}_c$. Below, we list some examples of graphs involved in this paper.

Table 4: Notations for graphs involved in this paper

| Symbols | Definitions |
|---|---|
| $\mathcal{G}$ | the graph space |
| $\mathcal{G}_c$ | the space of subgraphs with respect to the graphs from $\mathcal{G}$ |
| $\mathcal{Y}$ | the label space |
| $G \in \mathcal{G}$ | a graph |
| $G = (A, X)$ | a graph with the adjacency matrix $A \in \{0,1\}^{n \times n}$ and node feature matrix $X \in \mathbb{R}^{n \times d}$ |
| $\{G\}$ | a set of graphs |
| $G^p$ | a graph sampled as positive samples |
| $G^n$ | a graph sampled as negative samples |
| $G^s$ | a graph sampled according to CIGA [8] |
| $G_c$ | the invariant subgraph with respect to $G$ |
| $G_s$ | the spurious subgraph with respect to $G$ |
| $G_c^p$ | the invariant subgraph of a positive graph $G^p$ |
| $G_s^p$ | the spurious subgraph of a positive graph $G^p$ |
| $\widehat{G}_c$ | the estimated invariant subgraph |
| $\widehat{G}_s$ | the estimated spurious subgraph |
| $\widehat{G}_c^p$ | the estimated invariant subgraph of a positive graph $G^p$ |
| $\widehat{G}_s^p$ | the estimated spurious subgraph of a positive graph $G^p$ |
| $\triangle\widehat{G}_c \subseteq G_c$ | the part of the underlying invariant subgraph $G_c$ appeared in $\widehat{G}_c$ |
| $\overline{\triangle}\widehat{G}_c = G_c - \triangle\widehat{G}_c$ | the complementary part of $\triangle\widehat{G}_c$ with respect to the invariant subgraph $G_c$ |

# B   Limitations and Future Directions

Although our work establishes a set of minimal assumptions for feasible invariant graph learning when the environment partitions and auxiliary information about the environment are both not available, our work is built upon the minimal availability of the environment knowledge. Nevertheless, there could exist some additional information that may be helpful for environment augmentation. Therefore, it remains interesting to explore more theoretically grounded strategies to discover and leverage more environment information for identifying the graph invariance. When the direct environment augmentation is not feasible, GALA provides a suitable framework that one could easily manipulate the environment assistant model or the partitioning of the positive and negative graphs, to select the spurious features via the additional information and better identify the graph invariance.

In addition to the correlation strengths discussed in this work, there exist other factors, such as the size of spurious and invariant subgraphs, that affect the fitting of spurious and invariant patterns, another promising future direction is to discuss the influence of these factors to the design of environment assistant model and OOD generalization on graphs.

Besides, a better data partitioning strategy can be developed with uncertainty measures [51].

# C   Full Details of the Background

We give a more detailed background introduction about GNNs and Invariant Learning in this section.

**Graph Neural Networks.** Let $G = (A, X)$ denote a graph with $n$ nodes and $m$ edges, where $A \in \{0,1\}^{n \times n}$ is the adjacency matrix, and $X \in \mathbb{R}^{n \times d}$ is the node feature matrix with a node feature dimension of $d$. In graph classification, we are given a set of $N$ graphs $\{G_i\}_{i=1}^N \subseteq \mathcal{G}$ and their labels

$\{Y_i\}_{i=1}^N \subseteq \mathcal{Y} = \mathbb{R}^c$ from $c$ classes. Then, we train a GNN $\rho \circ h$ with an encoder $h : \mathcal{G} \to \mathbb{R}^h$ that learns a meaningful representation $h_G$ for each graph $G$ to help predict their labels $y_G = \rho(h_G)$ with a downstream classifier $\rho : \mathbb{R}^h \to \mathcal{Y}$. The representation $h_G$ is typically obtained by performing pooling with a READOUT function on the learned node representations:

$$h_G = \text{READOUT}(\{h_u^{(K)} | u \in V\}), \tag{10}$$

where the READOUT is a permutation invariant function (e.g., SUM, MEAN) [75], and $h_u^{(K)}$ stands for the node representation of $u \in V$ at $K$-th layer that is obtained by neighbor aggregation:

$$h_u^{(K)} = \sigma(W_K \cdot a(\{h_v^{(K-1)}\} | v \in \mathcal{N}(u) \cup \{u\})), \tag{11}$$

where $\mathcal{N}(u)$ is the set of neighbors of node $u$, $\sigma(\cdot)$ is an activation function, e.g., ReLU, and $a(\cdot)$ is an aggregation function over neighbors, e.g., MEAN.

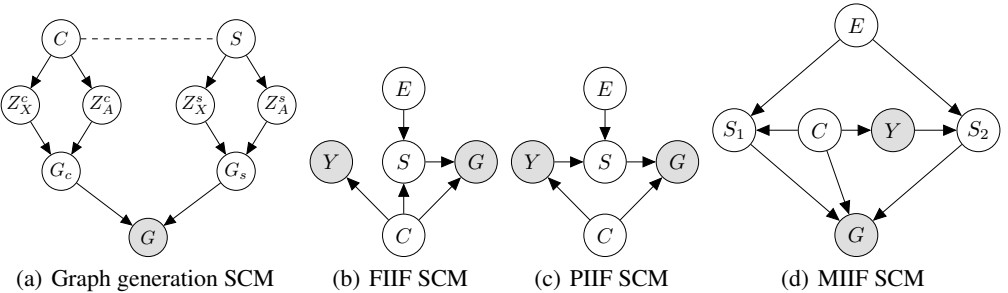

(a) Graph generation SCM     (b) FIIF SCM     (c) PIIF SCM     (d) MIIF SCM

Figure 4: Full SCMs on Graph Distribution Shifts [8].

**Graph generation process.**     This work focuses on graph classification, while the results generalize to node classification as well using the same setting as in Wu et al. [72]. Specifically, we are given a set of graph datasets $\mathcal{D} = \{\mathcal{D}_e\}_e$ collected from multiple environments $\mathcal{E}_{\text{all}}$. Samples $(G_i^e, Y_i^e) \in \mathcal{D}^e$ from the same environment are considered as drawn independently from an identical distribution $\mathbb{P}^e$. We consider the graph generation process proposed by Chen et al. [8] that covers a broad case of graph distribution shifts. Fig. 4 shows the full graph generation process considered in Chen et al. [8]. The generation of the observed graph $G$ and labels $Y$ are controlled by a set of latent causal variable $C$ and spurious variable $S$, i.e.,

$$G := f_{\text{gen}}(C, S).$$

$C$ and $S$ control the generation of $G$ by controlling the underlying invariant subgraph $G_c$ and spurious subgraph $G_s$, respectively. Since $S$ can be affected by the environment $E$, the correlation between $Y, S$ and $G_s$ can change arbitrarily when the environment changes. $C$ and $S$ control the generation of the underlying invariant subgraph $G_c$ and spurious subgraph $G_s$, respectively. Since $S$ can be affected by the environment $E$, the correlation between $Y, S$ and $G_s$ can change arbitrarily when the environment changes. Besides, the latent interaction among $C, S$ and $Y$ can be further categorized into *Full Informative Invariant Features* (*FIIF*) when $Y \perp\!\!\!\perp S|C$ and *Partially Informative Invariant Features* (*PIIF*) when $Y \not\perp\!\!\!\perp S|C$. Furthermore, PIIF and FIIF shifts can be mixed together and yield *Mixed Informative Invariant Features* (*MIIF*), as shown in Fig. 4. We refer interested readers to Chen et al. [8] for a detailed introduction of the graph generation process.

**Invariant graph representation learning.**     To tackle the OOD generalization challenge on graphs from Fig. 4, the existing invariant graph learning approaches generically aim to identify the underlying invariant subgraph $G_c$ to predict the label $Y$ [8, 72]. Specifically, the goal of OOD generalization on graphs is to learn an *invariant GNN* $f := f_c \circ g$, which is composed of two modules: a) a featurizer $g : \mathcal{G} \to \mathcal{G}_c$ that extracts the invariant subgraph $G_c$; b) a classifier $f_c : \mathcal{G}_c \to \mathcal{Y}$ that predicts the label $Y$ based on the extracted $G_c$, where $\mathcal{G}_c$ refers to the space of subgraphs of $\mathcal{G}$. The learning objectives of $f_c$ and $g$ are formulated as

$$\max_{f_c, g} I(\widehat{G}_c; Y), \text{ s.t. } \widehat{G}_c \perp\!\!\!\perp E, \widehat{G}_c = g(G). \tag{12}$$

Since $E$ is not observed, many strategies are proposed to impose the independence of $\widehat{G}_c$ and $E$. A common approach is to augment the environment information. For example, based on the

estimated invariant subgraphs $\widehat{G}_c$ and spurious subgraphs $\widehat{G}_s$, Liu et al. [45], Wu et al. [72, 73] proposed to generate new environments, while Li et al. [41], Yang et al. [77] proposed to infer the underlying environment labels. However, we show that it is fundamentally impossible to augment faithful environment information in Sec. 3. Miao et al. [55, 56], Yu et al. [81, 82, 83] adopt graph information bottleneck to tackle FIIF graph shifts, and they cannot generalize to PIIF shifts. Our work focuses on PIIF shifts, as it is more challenging when without environment labels [43]. Fan et al. [17] generalized [40] to tackle severe graph biases, i.e., when $H(S|Y) < H(C|Y)$. Chen et al. [8] proposed a contrastive framework to tackle both FIIF and PIFF graph shifts, but limited to $H(S|Y) > H(C|Y)$. However, in practice it is usually unknown whether $H(S|Y) < H(C|Y)$ or $H(S|Y) > H(C|Y)$ without environment information.

**More OOD generalization on graphs.** In addition to the aforementioned invariant learning approaches, Bevilacqua et al. [5], Li et al. [42], Yehudai et al. [78], Zhou et al. [89] study the OOD generalization as an extrapolation from small graphs to larger graphs in the task of graph classification and link prediction. In contrast, we study OOD generalization against various graph distribution shifts formulated in Fig. 4. In addition to the standard OOD generalization tasks studied in this paper, Mahdavi et al. [52], Xu et al. [76] study the OOD generalization in tasks of algorithmic reasoning on graphs. Jin et al. [29] study the test-time adaption in the graph regime. Kamhoua et al. [31] study the 3D shape matching under the presence of noises. Gui et al. [22] propose an independence constraint onto the target label and environment label to improve the OOD generalization when environment labels are available. Liu et al. [48] adopt a flexible framework to tackle shifting graph distributions. Chen et al. [7], Tao et al. [65], Zhou et al. [91, 92] study the OOD generalization on graphs from the adversarial robustness perspective.

In addition to graph classification, Chen et al. [6], Wu et al. [72] study node classification. Liu et al. [47] propose a structural reweighting strategy to improve the OOD generalization of node classification. Lee et al. [39] propose to incorporate multiple modules to handle different degree modes in OOD node classification. You et al. [80], Zhu et al. [93] study unsupervised graph domain adaption.Gao et al. [19], Zhou et al. [88, 90] study the OOD link prediction.

Besides, Lucic et al. [49] aims to find counterfactual subgraphs for explaining GNNs, which focuses on post-hoc explainability while this work focuses on intrinsic interpretability.

**Invariant learning without environment labels.** There are also plentiful studies in invariant learning without environment labels. Creager et al. [12] proposed a minmax formulation to infer the environment labels. Liu et al. [46] proposed a self-boosting framework based on the estimated invariant and variant features. Liu et al. [44], Zhang et al. [85] proposed to infer labels based the predictions of an ERM trained model. Deng et al. [14], Pezeshki et al. [61] improve the inference of group labels based on feature learning and prediction correctness. However, Lin et al. [43] found failure cases in Euclidean data where it is impossible to identify the invariant features without given environment labels. Moreover, as the OOD generalization on graphs is fundamentally more difficult than Euclidean data [8], the question about the feasibility of learning invariant subgraphs without environment labels remains unanswered.

# D   More Details about the Failure Cases

We provide more empirical results and details about the failure case verification experiments in complementary to Sec. 3. The results are shown in Fig. 5. We compared different environment augmentation approaches the vanilla GNN model trained with ERM (termed ERM), and an interpretable GNN model trained with ERM (termed XGNN).

The failure cases are constructed according to the two-piece graph generation models. The specific description is given as the following.

**Definition D.1** (3-class two-piece graphs). *Each environment is defined with two parameters, $\alpha_e, \beta_e \in [0, 1]$, and the dataset $\mathcal{D}_e$ is generated as follows:*

(a) *Sample $y^e \in \{0, 1, 2\}$ uniformly;*

(b) *Generate $G_c$ and $G_s$ via :*

$$G_c := f_{gen}^{G_c}(Y \cdot Rad(\alpha_e)), \; G_s := f_{gen}^{G_s}(Y \cdot Rad(\beta_e)),$$

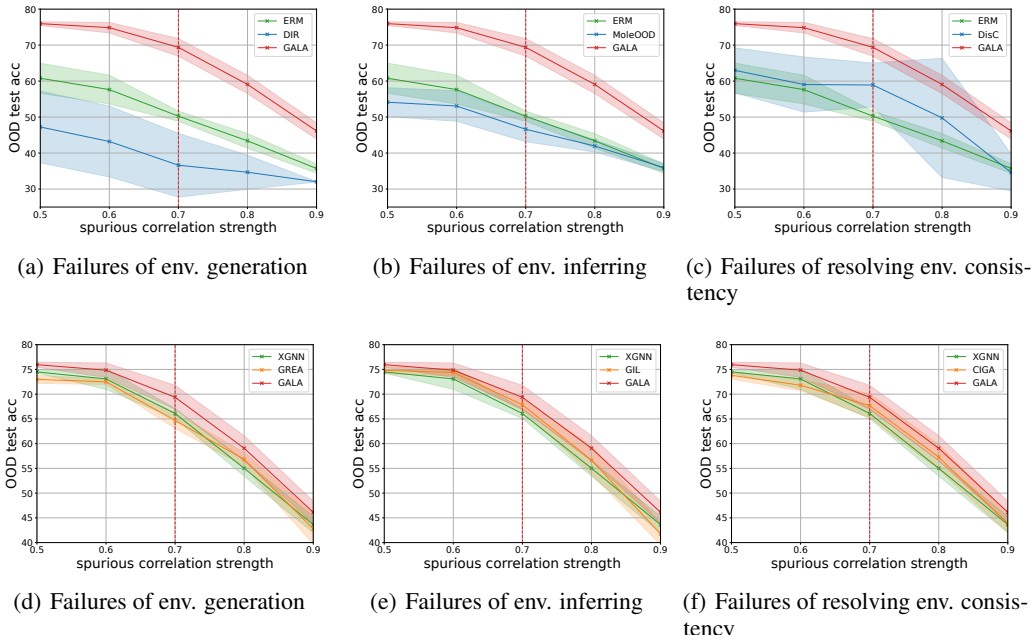

(a) Failures of env. generation  (b) Failures of env. inferring  (c) Failures of resolving env. consistency

(d) Failures of env. generation  (e) Failures of env. inferring  (f) Failures of resolving env. consistency

Figure 5: Failures of finding faithful environment information. Results shown in the figure are based on the 3 class two-piece graphs (Def. D.1), where the invariant correlation strength is fixed as $0.7$ while the spurious correlation strength is varied from $0.5$ to $0.7$. We can find that both environment augmentation and inferring approaches suffer from severe performance decreases or even underperform ERM and XGNN when the dominated correlation is not suitable for the method. In contrast, GALA maintains strong OOD performance for both cases.

*where $f_{gen}^{G_c}, f_{gen}^{G_s}$ respectively map input $\{0, 1, 2\}$ to a specific graph selected from a given set, and $Rad(\alpha)$ is a random variable with probability $\alpha$ taking a uniformly random value from $\{0, 1, 2\}$, and a probability of $1 - \alpha$ taking the value of $+1$;*

*(c) Sythesize $G$ by randomly concatenating $G_c$ and $G_s$:*

$$G := f_{gen}^G(G_c, G_s).$$

In experiments, we implement the 3-class two-piece graphs with the BA-motifs [50] model.

In experiments, we adopt a 3-layer GIN [75] with a hidden dimension of $32$ and a dropout rate of $0.0$ as the GNN encoder. The XGNN architecture is implemented via two GNNs following the original implementation as CIGA. The optimization is proceeded with Adam [34] using a learning rate of $1e - 3$. All experiments are repeated with 5 different random seeds of $\{1, 2, 3, 4, 5\}$. The mean and standard deviation are reported from the 5 runs.

We implement DIR [73], GREA [45], MoleOOD [77], GIL [41], DisC [17], and CIGA [8], according to the author provided codes (if available). As for the hyperparameters in each method, we use a penalty weight of $1e - 2$ for DIR following its original experiment in spurious motif datasets generated similarly using BA-motifs [73]. We use a penalty weight of $1$ for GREA as we empirically it does not affect the performance by changing to different weights. For MoleOOD and GIL, we set the number of environments as 3. We tune the penalty weights of MoleOOD with values from $\{1e - 2, 1e - 1, 1, 10\}$ but did not observe much performance differences. We tune the penalty weights of GIL with values from $\{1e - 5, 1e - 3, 1e - 1\}$ recommended by the authors. For DisC, we tune only the $q$ weight from $\{0.9, 0.7, 0.5\}$ in the GCE loss as we did not observe performance differences by changing the weight of the other term. We tune the penalty weight of CIGA with values from $\{0.5, 1, 2, 4, 8, 16, 32\}$ as recommended by the authors.

# E  Proofs for Theorems and Propositions

## E.1  Proof of Proposition 3.2

**Proposition E.1.** *(Restatement of Proposition 3.2) Consider the two-piece graph dataset $\mathcal{E}_{tr} = \{(\alpha, \beta_1), (\alpha, \beta_2)\}$ with $\alpha \geq \beta_1, \beta_2$ (e.g., $\mathcal{E}_{tr} = \{(0.25, 0.1), (0.25, 0.2)\}$), and its corresponding mixed environment $\mathcal{E}_{tr}^{mix} = \{\alpha, (\beta_1 + \beta_2)/2\}$ (e.g., $\mathcal{E}_{tr}^{mix} = \{(0.25, 0.15)\}$). When $\widehat{G}_c = G_s$ and $\widehat{G}_s = G_c$, it holds that the augmented environment $\mathcal{E}_v$ is also a two-piece graph dataset with*

$$\mathcal{E}_v = \{(0.5, (\beta_1 + \beta_2)/2)\} \text{ (e.g., } \mathcal{E}_v = \{(0.5, 0.15)\}).$$

*Proof.* From Definition 3.1, we known that for each graph $G_i \sim \mathcal{E}_{tr}^{mix} = \{(\alpha, (\beta_1 + \beta_2)/2)\}$, $G_i$ is the concatenation of the $G_c^i$ and $G_s^i$ defined as

$$G_c^i := f_{\text{gen}}^{G_c}(Y_i \cdot \text{Rad}(\alpha)_i), \quad G_s^i := f_{\text{gen}}^{G_s}(Y_i \cdot \text{Rad}((\beta_1 + \beta_2)/2)_i),$$

where $\text{Rad}(\cdot)_i$ denotes the $i$th sample of the random variable $\text{Rad}(\cdot)$.

Denote

$$G_A = f_{\text{gen}}^{G_c}(+1), \ G_B = f_{\text{gen}}^{G_c}(-1),$$

and

$$G_C = f_{\text{gen}}^{G_s}(+1), \ G_D = f_{\text{gen}}^{G_s}(-1),$$

Considering applying the augmentation to $2n$ samples randomly sampled from $\mathcal{E}_{tr}^{mix}$, since the featurizer $g$ separates each $G \in \mathcal{E}_{tr}^{mix}$ into $\widehat{G}_c = G_s$ and $\widehat{G}_s = G_c$, and the augmented graph $G^i$ is obtained by

$$G^{i,j} = f_{\text{gen}}^{G}(\widehat{G}_c^i, \widehat{G}_s^j), \forall i, j \in \{1...n\}.$$

Then, the new $\alpha_v, \beta_v$ in $\mathcal{E}_v$ can be obtained by summing up the overall numbers of $G_A, G_B, G_C, G_D$ concatenated into $2n^2$ samples in $\mathcal{E}_v$.

Specifically, we can inspect the changes of the distributions of motifs and labels. Let $\bar{\beta} = (\beta_1 + \beta_2)/2$, without loss of generality, we focus on inspecting the changes given $Y = +1$, since the changes given $Y = -1$ is symmetric as $Y = +1$. The original distribution is shown as follows:

| $Y = +1$ | $G_A$ | $G_B$ |
|---|---|---|
| $G_C$ | $(1 - \alpha)(1 - \bar{\beta})n$ | $\alpha(1 - \bar{\beta})n$ |
| $G_D$ | $(1 - \alpha)\bar{\beta}n$ | $\alpha\bar{\beta}n$ |

Then, new distributions of the motifs and labels are determined by the number of original motifs identified as $\widehat{G}_c$ and $\widehat{G}_s$, respectively. When $\widehat{G}_c = G_s$ and $\widehat{G}_s = G_c$, in the new environment $\mathcal{E}_v$, given $Y = +1$, $G_C$ contributes $(1 - \bar{\beta})n * 2n$ samples as the "invariant" subgraph. More specifically, $G_C$ will be concatenated with $G_A$ and $G_B$ by $n$ times, respectively. Then we have the new distribution tables shown as follows:

| $Y = +1$ | $G_A$ | $G_B$ |
|---|---|---|
| $G_C$ | $(1 - \bar{\beta})n^2$ | $(1 - \bar{\beta})n^2$ |
| $G_D$ | $\bar{\beta}n^2$ | $\bar{\beta}n^2$ |

Since given the same $Y$, the spurious subgraph $G_C$ and $G_D$ will still have the same chance being flipped, we have $\beta_v = \bar{\beta}$. While as $G_A$ and $G_B$ appear the same times given the same $Y$, it suffices to know that $\alpha_v = 0.5$. □

## E.2  Complementary discussion for Sec. 3.1

**Proposition E.2.** *Given the same graph generation process as in Fig. 2, when there exists spurious subgraph $G_s$ such that $P^{e_1}(Y|G_s) = P^{e_2}(Y|G_s)$ for any two environments $e_1, e_2 \in \mathcal{E}_{tr}$, where $P^e(Y|G_s)$ is the conditional distribution $P(Y|G_s)$ under environment $e \in \mathcal{E}_{all}$, it is impossible for any learning algorithm applied to $f_c \circ g$ to differentiate $G_c$ from $G_s$.*

*Proof.* Let $G_s^*$ be the spurious subgraph such that $P^{e_1}(Y|G_s) = P^{e_2}(Y|G_s)$ for any two environments $e_1, e_2 \in \mathcal{E}_{tr}$, and $G_c$ be the invariant subgraph which $P^{e_1}(Y|G_c) = P^{e_2}(Y|G_c)$, $\forall e_1, e_2 \in \mathcal{E}_{tr}$ by definition. Consider a learning algorithm applied to $f_c \circ g$ that accepts the input of $\mathcal{E}_{tr}^{mix}$, and extracts a subgraph $\widehat{G}_c = g(Y)$ as an estimation of the invariant subgraph for any $G$ to predict $Y$ via $f_c(\widehat{G}_c)$ in a deterministic manner. If the algorithm succeed to extract $G_c$ from $\mathcal{E}_{tr}^{mix}$, then there always exists a $\mathcal{E}_{tr}^{mix'}$ with the desired spurious subgraph $G_s'$ and a underlying invariant subgraph $G_c'$, such that $G_s' = G_c$ and $G_c' = G_s^*$. Due to the deterministic nature, the algorithm fails to identify $G_c'$ in $\mathcal{E}_{tr}^{mix'}$. $\qquad\square$

### E.3 Proof of Proposition 3.4

**Proposition E.3.** *(Restatement of Proposition 3.4) There exist* 2 *two-piece graph training environments* $\mathcal{E}_{tr}$ *and* $\mathcal{E}_{tr}'$ *that share the same joint distribution* $P(Y, G)$. *Any learning algorithm will fail in either* $\mathcal{E}_{tr}$ *or* $\mathcal{E}_{tr}'$.

*Proof.* Let the mixed training environment of $\mathcal{E}_{tr}$ and $\mathcal{E}_{tr}'$ be $\mathcal{E}_{tr}^{mix} = \{(\alpha, \beta)\}$. Based on the definition of two-piece graphs (Definition 3.1), the joint distribution of the mixed training dataset $(G = \text{Concat}[G_c, G_s], Y)$ can be computed as

$$
\begin{cases}
Y = +1, & \text{with probability } 0.5, \\
Y = -1, & \text{with probability } 0.5, \\
\text{Bit}^{G_c}(G_c) = \text{Bit}^{G_s}(G_s) = Y, & \text{with probability } (1-\alpha)(1-\beta), \\
\text{Bit}^{G_c}(G_c) \neq \text{Bit}^{G_s}(G_s) = Y, & \text{with probability } \alpha(1-\beta), \\
\text{Bit}^{G_s}(G_s) \neq \text{Bit}^{G_c}(G_c) = Y, & \text{with probability } (1-\alpha)\beta, \\
\text{Bit}^{G_c}(G_c) = \text{Bit}^{G_s}(G_s) \neq Y, & \text{with probability } \alpha\beta.
\end{cases}
$$

Here we use $\text{Bit}^{G_c}(G_c)$ to obtain the input bit of a subgraph $G_c$ (or $(f_{gen}^{G_c})^{-1}$), and $\text{Bit}^{G_s}(G_s)$ for $G_s$, respectively.

Any learning algorithm that tries to identify the invariant subgraph from this training dataset will compute a model that uses subgraph $G_c$, or subgraph $G_s$, or both $G_c$ and $G_s$ to predict $Y$ deterministically. Thus, as long as the joint distribution does not change, the resulting model will always identify the same invariant subgraph. Without loss of generality, let us assume that the model correctly identifies $G_c$ as the invariant subgraph for $\mathcal{E}_{tr} = \{(\alpha, \beta_1), (\alpha, \beta_2)\}$ with $\beta = (\beta_1 + \beta_2)/2$.

Now let the other training environment be $\mathcal{E}_{tr}' = \{(\alpha_1, \beta), (\alpha_2, \beta)\}$ with $\alpha = (\alpha_1 + \alpha_2)/2$. It is clear that since the mixed training environment of $\mathcal{E}_{tr}'$ is still $\{(\alpha, \beta)\}$, the model keeps regarding $G_c$ as the invariant subgraph. However, for $\mathcal{E}_{tr}'$, the model fails to identify the invariance since now the invariant subgraph is $G_s$.

$\qquad\square$

### E.4 Proof of Corollary 3.6

**Corollary E.4.** *(Restatement of Corollary 3.6) Without Assumption 3.3 or Assumption 3.5, there does not exist a learning algorithm that captures the invariance of the two-piece graph environments.*

*Proof.* The proof for lacking Assumption 3.3 is identical to the proof for Proposition E.2. Consider a learning algorithm applied to $f_c \circ g$ that accepts the input of $\mathcal{E}_{tr}^{mix}$, and extracts a subgraph $\widehat{G}_c = g(Y)$ as an estimation of the invariant subgraph for any $G$ to predict $Y$ via $f_c(\widehat{G}_c)$ in a deterministic manner. Without the holding of Assumption 3.5, due to Proposition 3.4, there exists $\mathcal{E}_{tr}^{mix'}$ for each $\mathcal{E}_{tr}^{mix}$ that have the identical joint distribution but different underlying invariant subgraph. Thus, any learning algorithm that succeeds in either $\mathcal{E}_{tr}^{mix}$ or $\mathcal{E}_{tr}^{mix'}$ will fail in the other. $\qquad\square$

### E.5 Proof of Theorem 4.1

**Theorem E.5.** *(Restatement of Theorem 4.1) Given, i) the same data generation process as in Fig. 2; ii)* $\mathcal{D}_{tr}$ *that satisfies variation sufficiency (Assumption 3.3) and variation consistency*

*(Assumption 3.5); iii) $\{G^p\}$ and $\{G^n\}$ are distinct subsets of $\mathcal{D}_{\text{tr}}$ such that $I(G_s^p; G_s^n|Y) = 0$, $\forall G_s^p = \arg\max_{\widehat{G}_s^p} I(\widehat{G}_s^p; Y)$ under $\{G^p\}$, and $\forall G_s^n = \arg\max_{\widehat{G}_s^n} I(\widehat{G}_s^n; Y)$ under $\{G^n\}$; suppose $|G_c| = s_c$, $\forall G_c$, resolving the following $\mathtt{GALA}$ objective elicits an invariant GNN defined via Eq. 12,*

$$\max_{f_c, g} \ I(\widehat{G}_c; Y), \ s.t. \ g \in \arg\max_{\hat{g}, |\widehat{G}_c^p| \leq s_c} I(\widehat{G}_c^p; \widehat{G}_c^n|Y), \tag{13}$$

*where $\widehat{G}_c^p \in \{\widehat{G}_c^p = g(G^p)\}$ and $\widehat{G}_c^n \in \{\widehat{G}_c^n = g(G^n)\}$ are the estimated invariant subgraphs via $g$ from $\{G^p\}$ and $\{G^n\}$, respectively.*

*Proof.* Without loss of generality, we assume that $\{G^p\}$ has the same spurious dominance situation as $\mathcal{E}_{\text{tr}}$. In other words, when $H(S|Y) < H(C|Y)$, the data distribution in $\{G^p\}$ also follows $H(S|Y) < H(C|Y)$, while $H(S|Y) > H(C|Y)$ in $\{G^n\}$. To proceed, we will use the language of Chen et al. [8].

We begin by discussing the case of $H(S|Y) < H(C|Y)$. Given $H(S|Y) < H(C|Y)$, we have $H(S|Y) < H(C|Y)$ in $\{G^p\}$ and $H(S|Y) > H(C|Y)$ in $\{G^n\}$. Then, we claim that

$$G_c \in \arg\max_{\widehat{G}_c^p, |\widehat{G}_c^p| \leq s_c} I(\widehat{G}_c^p; \widehat{G}_c^n|Y). \tag{14}$$

Otherwise, consider there exists a subgraph of the spurious subgraph $\triangle\widehat{G}_s^p \subseteq G_s^p$ in $\widehat{G}_c^p$, which takes up the space of $\triangle\widehat{G}_c^p \subseteq G_c^p$ from $\widehat{G}_c^p$. Then, let $\overline{\triangle}\widehat{G}_c^p = G_c^p - \triangle\widehat{G}_c^p$ we can inspect the changes to $I(\widehat{G}_c^p; \widehat{G}_c^n|Y)$ led by $\triangle\widehat{G}_s^p$:

$$\triangle I(\widehat{G}_c^p; \widehat{G}_c^n|Y)$$
$$= \triangle H(\widehat{G}_c^p|Y) - \triangle H(\widehat{G}_c^p|\widehat{G}_c^n, Y)$$
$$= \left[ H(\overline{\triangle}\widehat{G}_c^p, \triangle\widehat{G}_s^p|Y) - H(\overline{\triangle}\widehat{G}_c^p, \triangle\widehat{G}_c^p|Y) \right] - \left[ H(\overline{\triangle}\widehat{G}_c^p, \triangle\widehat{G}_s^p|\widehat{G}_c^n, Y) - H(\overline{\triangle}\widehat{G}_c^p, \triangle\widehat{G}_c^p|\widehat{G}_c^n, Y) \right]$$
$$= \left[ H(\triangle\widehat{G}_s^p|\overline{\triangle}\widehat{G}_c^p, Y) - H(\triangle\widehat{G}_c^p|\overline{\triangle}\widehat{G}_c^p, Y) \right] - \left[ H(\triangle\widehat{G}_s^p|\overline{\triangle}\widehat{G}_c^p, \widehat{G}_c^n, Y) - H(\triangle\widehat{G}_c^p|\overline{\triangle}\widehat{G}_c^p, \widehat{G}_c^n, Y) \right], \tag{15}$$

where the last equality is obtained via expanding the conditional entropy. Then, considering the contents in $\widehat{G}_c^n$, without loss of generality, we can divide all of the possible cases into two:

(i) $\widehat{G}_c^n$ contains only the corresponding invariant subgraph $G_c^n$;

(ii) $\widehat{G}_c^n$ contains subgraph from the corresponding spurious subgraph $G_s^n$, denoted as $\triangle\widehat{G}_s^n \subseteq G_s^n$;

For case (i), it is easy to write Eq. 15 as:

$$\triangle I(\widehat{G}_c^p; \widehat{G}_c^n|Y)$$
$$= \left[ H(\triangle\widehat{G}_s^p|\overline{\triangle}\widehat{G}_c^p, Y) - H(\triangle\widehat{G}_c^p|\overline{\triangle}\widehat{G}_c^p, Y) \right] - \left[ H(\triangle\widehat{G}_s^p|\overline{\triangle}\widehat{G}_c^p, \widehat{G}_c^n, Y) - H(\triangle\widehat{G}_c^p|\overline{\triangle}\widehat{G}_c^p, \widehat{G}_c^n, Y) \right],$$
$$= -H(\triangle\widehat{G}_c^p|\overline{\triangle}\widehat{G}_c^p, Y) + H(\widehat{G}_c^p|\overline{\triangle}\widehat{G}_c^p, \widehat{G}_c^n, Y), \tag{16}$$

since $H(\triangle\widehat{G}_s^p|\overline{\triangle}\widehat{G}_c^p, Y) = H(\triangle\widehat{G}_s^p|\widehat{G}_c^n, \overline{\triangle}\widehat{G}_c^p, Y) = H(\triangle\widehat{G}_s^p|Y)$ given $C \perp\!\!\!\perp S|Y$ for PIIF shifts. Then, it suffices to know that $\triangle I(\widehat{G}_c^p; \widehat{G}_c^n|Y) \leq 0$ as conditioning on new variables will not increase the entropy [79].

For case (ii), we have :

$$\triangle I(\widehat{G}_c^p; \widehat{G}_c^n|Y)$$
$$= \left[ H(\triangle\widehat{G}_s^p|\overline{\triangle}\widehat{G}_c^p, Y) - H(\triangle\widehat{G}_c^p|\overline{\triangle}\widehat{G}_c^p, Y) \right] - \left[ H(\triangle\widehat{G}_s^p|\overline{\triangle}\widehat{G}_c^p, \widehat{G}_c^n, Y) - H(\triangle\widehat{G}_c^p|\overline{\triangle}\widehat{G}_c^p, \widehat{G}_c^n, Y) \right],$$
$$= \left[ -H(\triangle\widehat{G}_c^p|\overline{\triangle}\widehat{G}_c^p, Y) + H(\triangle\widehat{G}_c^p|\overline{\triangle}\widehat{G}_c^p, \widehat{G}_c^n, Y) \right] + \left[ H(\triangle\widehat{G}_s^p|\overline{\triangle}\widehat{G}_c^p, Y) - H(\triangle\widehat{G}_s^p|\overline{\triangle}\widehat{G}_c^p, \widehat{G}_c^n, Y) \right], \tag{17}$$

where we claim that $H(\triangle\widehat{G}_s^p|\overline{\triangle}\widehat{G}_c^p, Y) - H(\triangle\widehat{G}_s^p|\overline{\triangle}\widehat{G}_c^p, \widehat{G}_c^n, Y) = 0$, and similarly conclude that $\triangle I(\widehat{G}_c^p; \widehat{G}_c^n|Y) \leq 0$. More specifically, we can rewrite the first term in Eq. 17 as

$$H(\triangle\widehat{G}_s^p|\overline{\triangle}\widehat{G}_c^p, Y) - H(\triangle\widehat{G}_s^p|\overline{\triangle}\widehat{G}_c^p, \widehat{G}_c^n, Y) = H(\triangle\widehat{G}_s^p|Y) - H(\triangle\widehat{G}_s^p|\triangle\widehat{G}_s^n, Y)$$
$$= I(\triangle\widehat{G}_s^p; \triangle\widehat{G}_s^n|Y) = 0,$$

using the variation condition (i.e., assumption iii)) for $\triangle\widehat{G}_s^p$ under $\{G^p\}$, and $\triangle\widehat{G}_s^n$ under $\{G^n\}$.

After showing the success of GALA in tackling $H(S|Y) < H(C|Y)$, it also suffices to know that the aforementioned discussion also generalizes to the other case, i.e., when $H(S|Y) > H(C|Y)$ in $\{G^p\}$ and $H(S|Y) < H(C|Y)$ in $\{G^n\}$. $\qquad\square$

# F   More Discussions on Practical Implementations of GALA

In this section, we provide more implementation discussions about `GALA` in complementary to Sec. 4.

**Objective implementation.**   As the estimation of mutual information could be highly expensive [4, 67], inspired by Chen et al. [8], we adopt the contrastive learning to approximates the mutual information between subgraphs in Eq. 9 [4, 11, 33, 62, 67]:

$$
I(\widehat{G}_c^p; \widehat{G}_c^n|Y) \approx \mathbb{E}_{\substack{\{\widehat{G}_c^p, \widehat{G}_c^n\} \sim \mathbb{P}_g(G|\mathcal{Y}=Y) \\ \{G_c^i\}_{i=1}^M \sim \mathbb{P}_g(G|\mathcal{Y}\neq Y)}}
\log \frac{e^{\phi(h_{\widehat{G}_c^p}, h_{\widehat{G}_c^n})}}{e^{\phi(h_{\widehat{G}_c^p}, h_{\widehat{G}_c^n})} + \sum_{i=1}^M e^{\phi(h_{\widehat{G}_c}, h_{G_c^i})}}, \tag{18}
$$

where $(\widehat{G}_c^p, \widehat{G}_c^n)$ are subgraphs extracted by $g$ from $\{G^p\}, \{G^n\}$ that share the same label, respectively. $\{G_c^i\}_{i=1}^M$ are subgraphs extracted by $g$ from $G$ that has a different label. $\mathbb{P}_g(G|\mathcal{Y}=Y)$ is the push-forward distribution of $\mathbb{P}(G|\mathcal{Y}=Y)$ by featurizer $g$, $\mathbb{P}(G|\mathcal{Y}=Y)$ refers to the distribution of $G$ given the label $Y$, $\mathbb{P}(G|\mathcal{Y}\neq Y)$ refers to the distribution of $G$ given the label that is different from $Y$, $\widehat{G}_c = g(\widehat{G}), \widehat{G}_c = g(\widehat{G}), G_c^i = g(G^i)$ are the estimated subgraphs, $h_{\widehat{G}_c^p}, h_{\widehat{G}_c^n}, h_{G_c^i}$ are the graph presentations of the extracted subgraphs. $\phi$ is a similarity measure. As $M \to \infty$, Eq. 18 approximates $I(\widehat{G}_c^p; \widehat{G}_c^n|Y)$ [1, 32, 69].

**Environment assistant implementation.**   Theorem 4.1 shows the effectiveness of `GALA` when given proper subsets of $\{G^p\}$ and $\{G^n\}$. In practice, we can implement the environment assistant into multiple forms. As discussed in Sec. 4.1, ERM trained model can serve as a reliable proxy. Since ERM tends to learn the first dominant features, when $H(S|Y) < H(C|Y)$, ERM will firstly learn to extract spurious subgraphs $G_s$ to make predictions. Therefore, we can obtain $\{G^p\}$ by finding samples where ERM correctly predicts the labels, while $\{G^n\}$ for samples that ERM predicts an incorrect label. In addition to direct label predictions, we can also adopt clustering [85] to yield environment assistant predictions for better contrastive sampling. We provide the detailed description of the clustering based variant of `GALA` in Algorithm 2.

---

**Algorithm 2 GALA**: Clustering based **G**raph inv**A**riant **L**earning **A**ssistant

---

1: **Input:** Training data $\mathcal{D}_{\text{tr}}$; environment assistant $A$; featurizer $g$; classifier $f_c$; length of maximum training epochs $e$; batch size $b$;
2: Initialize environment assistant $A$;
3: **for** $p \in [1, \dots, e]$ **do**
4:      Sample a batch of data $\{G_i, Y_i\}_{i=1}^b$ from $\mathcal{D}_{\text{tr}}$;
5:      Obtain Environment Assistant predictions $\{\hat{c}_i^e\}_{i=1}^b$ using $k$-means clustering on the graph representations yielded by $A$;
6:      **for** each sample $G_i, y_i \in \{G_i, Y_i\}_{i=1}^b$ **do**
7:          Find *postive* graphs with same $y_i$ and different $\hat{c}_i^e$;
8:          Find *negative* graphs with different $y_i$ but same environment assistant prediction $\hat{c}_i^e$;
9:          Calculate `GALA` risk via Eq. 18;
10:          Update $f_c, g$ via gradients from `GALA` risk;
11:      **end for**
12: **end for**
13: **return** final model $f_c \circ g$;

---

Empirically, we find clustering based variants can provide better performance when the spurious correlations are well learned by the environment assistant model. More concretely, we plot the umap visualizations [53] of ERM trained environment assistant model as in Fig. 6, where we can find that clustering predictions provide better approximations to the underlying group labels.

Besides, we can also incorporate models that are easier to overfit to the first dominant features to better differentiate $\{G^p\}$ from $\{G^n\}$. To demonstrate the influence of different environment assistant implementations, we conduct more studies with interpretable GNNs with an interpretable ratio of 30% trained with ERM and also with a CIGAv1 penalty of 4.

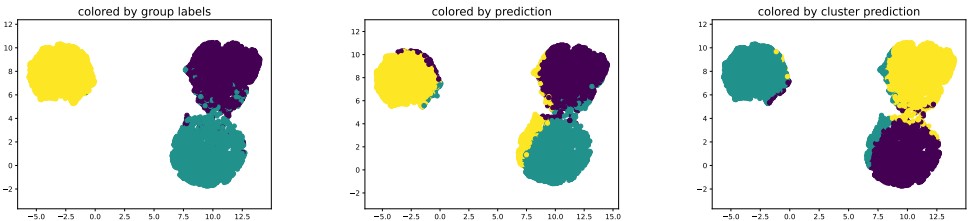

(a) Colored by environment labels.    (b) Colored by label predictions.    (c) Colored by cluster predictions.

Figure 6:  Umap visualizations of learned graph representations in ERM trained environment assistant model based on the 3-class two-piece graph $\{0.7, 0.9\}$.

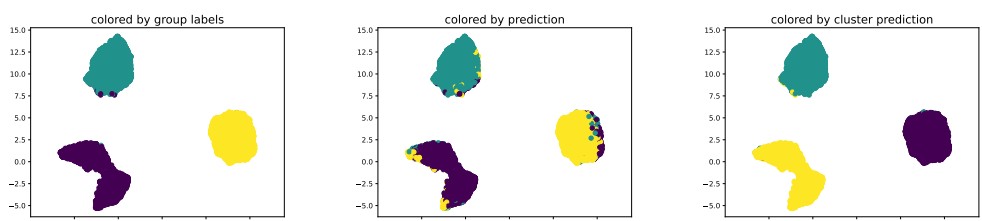

(a) Colored by environment labels.    (b) Colored by label predictions.    (c) Colored by cluster predictions.

Figure 7:  Umap visualizations of learned graph representations in an interpretable GNN model (ratio=30%) trained with ERM based on the 3-class two-piece graph $\{0.7, 0.9\}$.

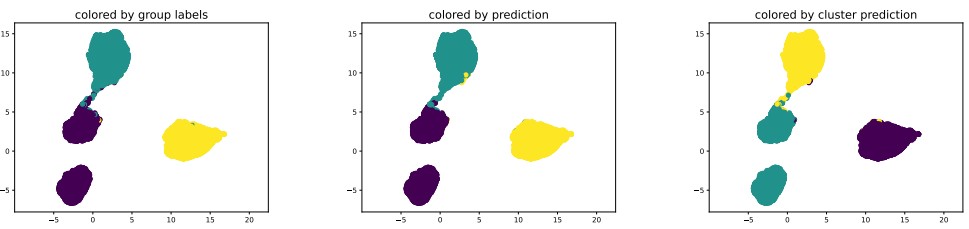

(a) Colored by environment labels.    (b) Colored by label predictions.    (c) Colored by cluster predictions.

Figure 8:  Umap visualizations of learned graph representations in an interpretable GNN model (ratio=30%) trained with ERM based on the 3-class two-piece graph $\{0.7, 0.9\}$.

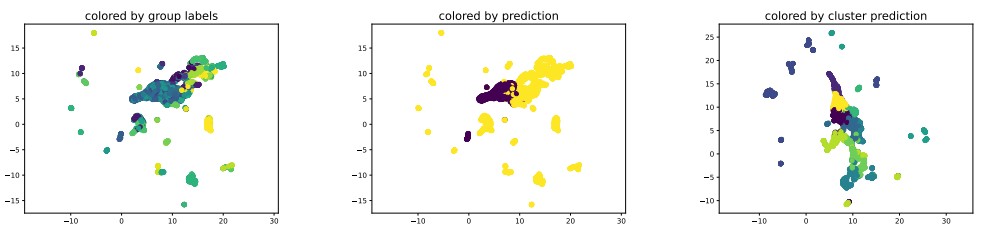

(a) Colored by environment labels.    (b) Colored by label predictions.    (c) Colored by cluster predictions.

Figure 9:  Umap visualizations of learned graph representations of a interpretable GNN trained by ERM on EC50-Assay.

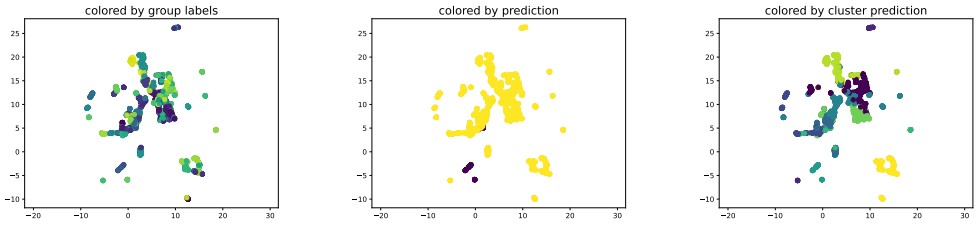

(a) Colored by environment labels.    (b) Colored by label predictions.    (c) Colored by cluster predictions.

Figure 10: Umap visualizations of learned graph representations of an interpretable GNN trained by ERM on EC50-Scaffold.

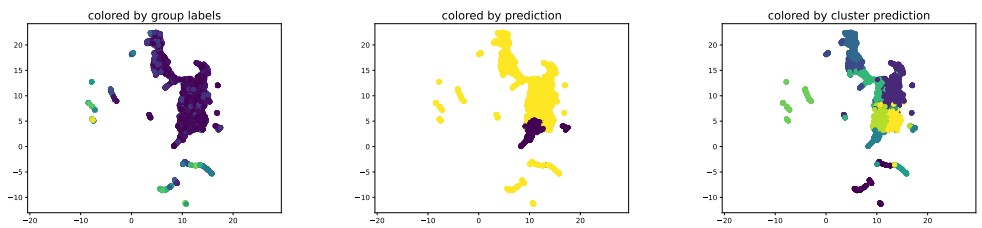

(a) Colored by environment labels.    (b) Colored by label predictions.    (c) Colored by cluster predictions.

Figure 11: Umap visualizations of learned graph representations of a interpretable GNN trained by ERM on EC50-Size.

In Fig. 7 and Fig. 8, it can be found that the interpretable GNN learns hidden representations that are better clustered with group labels. The clustering based predictions yields a better approximation of the underlying environment labels. Furthermore, when implementing the environment assistant model using a interpretable GNN as well as a CIGAv1 penalty, which facilitates the overfitting to the spurious correlations, then the vanilla label predictions can also yield a good approximation of the underlying environment labels.

Although using the clustering predictions seem to be promising, we also find negative cases. For example, in DrugOOD datasets, the number of curated environment labels are much larger that learning a well clusterd hidden representations for the environment labels appears to be difficult. Shown as in Fig. 9 to Fig. 11, the learned representations have poor quality for approximating the underlying environment labels. Empirically, we also find that direct using label predictions in DrugOOD datasets generically yield better performance.

**One-side contrastive sampling.** The original supervised contrastive implementation [33] takes positive and negative samples within the batch using two-side contrastive sampling. That is, all the samples will be considered as anchor points. However, when it is used to contrast samples from $\widehat{G}_c^p$ and $\widetilde{G}_c^n$, there could be undesired behaviors. First, it can often happen that there are few to no negative cases when the spurious correlations are too strong. The samples from $\{G^p\}$ in a batch may pull the representations of samples from $\{G^n\}$ to even closer, which makes the model further overfitted to the spurious correlations. Second, the sampling over $\widehat{G}_c^p$ and $\widetilde{G}_c^n$, can be seen as hard positive and negative samples, that may impose a too strong regularizations that preventing the learning of any correlations. Therefore, we propose to use one-side sampling. That is, only using the incorrectly predicted samples as anchor points. We empirically observe one-side sampling could yield better performance in two-piece graphs.

**Upsampling of minority group samples.** It is possible that the number of positive and negative graphs is imbalanced, especially when adopting the label predictions to sample positive and negative graphs. For example, when the ERM trained assistant model overfits the training distribution under the spuriousness-dominated case, the number of negative graphs will be extremely small. Given an

extremely small number of negative samples for contrastive learning, the resulting mutual information estimation will be collapsed to trivial solutions. Therefore, we propose a simple strategy to mitigate the issue. We directly upsample the minority group samples. The minority group of samples will be repeated $k$ times within the training set.

# G  More Details about the Experiments

In this section, we provide more details about the experiments, including the dataset preparation, baseline implementations, models and hyperparameters selection as well as the evaluation protocols.

## G.1  Datasets

We provide more details about the motivation and construction method of the datasets that are used in our experiments. Statistics of the datasets are presented in Table 5.

Table 5: Information about the datasets used in experiments. The number of nodes and edges are respectively taking average among all graphs.

| Datasets | # Training | # Validation | # Testing | # Classes | # Nodes | # Edges | Metrics |
|---|---|---|---|---|---|---|---|
| Two-piece graphs $\{0.8, 0.6\}$ | $9,000$ | $3,000$ | $3,000$ | 3 | 26.14 | 36.21 | ACC |
| Two-piece graphs $\{0.8, 0.7\}$ | $9,000$ | $3,000$ | $3,000$ | 3 | 26.18 | 36.27 | ACC |
| Two-piece graphs $\{0.8, 0.9\}$ | $9,000$ | $3,000$ | $3,000$ | 3 | 26.13 | 36.22 | ACC |
| Two-piece graphs $\{0.7, 0.9\}$ | $9,000$ | $3,000$ | $3,000$ | 3 | 26.13 | 36.22 | ACC |
| CMNIST-sp | $40,000$ | $5,000$ | $15,000$ | 2 | 56.90 | 373.85 | ACC |
| Graph-SST2 | $24,881$ | $7,004$ | $12,893$ | 2 | 10.20 | 18.40 | ACC |
| EC50-Assay | $4,978$ | $2,761$ | $2,725$ | 2 | 40.89 | 87.18 | ROC-AUC |
| EC50-Scaffold | $2,743$ | $2,723$ | $2,762$ | 2 | 35.54 | 75.56 | ROC-AUC |
| EC50-Size | $5,189$ | $2,495$ | $2,505$ | 2 | 35.12 | 75.30 | ROC-AUC |
| Ki-Assay | $8,490$ | $4,741$ | $4,720$ | 2 | 32.66 | 71.38 | ROC-AUC |
| Ki-Scaffold | $5,389$ | $4,805$ | $4,463$ | 2 | 29.96 | 65.11 | ROC-AUC |
| Ki-Size | $8,605$ | $4,486$ | $4,558$ | 2 | 30.35 | 66.49 | ROC-AUC |

**Two-piece graph datasets.** We construct 3-class synthetic datasets based on BAMotif [50] following Def. D.1, where the model needs to tell which one of three motifs (House, Cycle, Crane) the graph contains. For each dataset, we generate 3000 graphs for each class at the training set, 1000 graphs for each class at the validation set and testing set, respectively. Each dataset is defined with two variables $\{a, b\}$ referring to the strength of invariant and spurious correlations. Given $\{a, b\}$, we generate the training data following the percise generation process as Def. D.1. While for the generation of validation sets, we use a $b_v = \max(1/3, b - 0.2)$ that facilitates the model selection for OOD generalization [9, 23]. While for the generation of test datasets, we merely use a $b = 0.33$ that contains no distribution shifts, to fully examine to what extent the model learns the invariant correlations. During the construction, we merely inject the distribution shifts in the training data while keeping the testing data and validation data without the biases.

**CMNIST-sp.** To study the effects of PIIF shifts, we select the ColoredMNIST dataset created in IRM [3]. We convert the ColoredMnist into graphs using the superpixel algorithm introduced by Knyazev et al. [36]. Specifically, the original Mnist dataset is assigned to binary labels where images with digits $0 - 4$ are assigned to $y = 0$ and those with digits $5 - 9$ are assigned to $y = 1$. Then, $y$ will be flipped with a probability of $0.25$. Thirdly, green and red colors will be respectively assigned to images with labels $0$ and $1$ an averaged probability of $0.15$ (since we do not have environment splits) for the training data. While for the validation and testing data, the probability is flipped to $0.9$.

**Graph-SST2.** Inspired by the data splits generation for studying distribution shifts on graph sizes, we split the data curated from sentiment graph data [84], that converts sentiment sentence classification datasets **Graph-SST2** [63] into graphs, where node features are generated using BERT [15] and the edges are parsed by a Biaffine parser [20]. Our splits are created according to the averaged degrees of each graph. Specifically, we assign the graphs as follows: Those that have smaller or equal to 50-th percentile averaged degree are assigned to training, those that have averaged degree large than

50-th percentile while smaller than 80-th percentile are assigned to the validation set, and the left are assigned to test set.

**DrugOOD datasets.** To evaluate the OOD performance in realistic scenarios with realistic distribution shifts, we also include three datasets from DrugOOD benchmark [28]. DrugOOD is a systematic OOD benchmark for AI-aided drug discovery, focusing on the task of drug target binding affinity prediction for both macromolecule (protein target) and small-molecule (drug compound). The molecule data and the notations are curated from realistic ChEMBL database [54]. Complicated distribution shifts can happen on different assays, scaffolds and molecule sizes. In particular, we select `DrugOOD-lbap-core-ec50-assay`, `DrugOOD-lbap-core-ec50-scaffold`, `DrugOOD-lbap-core-ec50-size`, `DrugOOD-lbap-core-ki-assay`, `DrugOOD-lbap-core-ki-scaffold`, and `DrugOOD-lbap-core-ki-size`, from the task of Ligand Based Affinity Prediction which uses `ic50` measurement type and contains `core` level annotation noises. We directly use the data files provided by the authors.[4] For more details, we refer interested readers to Ji et al. [28].

### G.2   Baselines and Evaluation Setup

During the experiments, we do not tune the hyperparameters exhaustively while following the common recipes for optimizing GNNs. Details are as follows.

**GNN encoder.** For a fair comparison, we use the same GNN architecture as graph encoders for all methods. By default, we use 3-layer GIN [75] with Batch Normalization [27] between layers and JK residual connections at the last layer [74]. The hidden dimension is set to 32 for Two-piece graphs, CMNIST-sp, and 128 for SST2, and DrugOOD datasets. The pooling is by default a mean function over all nodes. The only exception is DrugOOD datasets, where we follow the backbone used in the paper [28], i.e., 4-layer GIN with sum readout.

**Interpretable GNN backbone.** As mentioned in Sec. 2 that most of the existing invariant graph learning approaches adopt the interpretable GNN as the basic backbone model for the whole predictor $f = f_c \circ g$, where $g : \mathcal{G} \to \mathcal{G}_c$ is a featurizer GNN and $f_c : \mathcal{G}_c \to \mathcal{Y}$ is a classifier GNN. $g$ first calculates the sampling weights as in $\widehat{G}_c$ for each edge. More formally, given a graph $G$ containing $n$ nodes, a soft mask is predicted through the following equation:

$$Z = \text{GNN}(G) \in \mathbb{R}^{n \times h}, \; M = \mathrm{a}(Z, A) \in \mathbb{R}^{n \times n},$$

where $a$ calculates the sampling weights for each edge using a MLP: $M_{ij} = \text{MLP}([Z_i, Z_j])$. Based on the continuous sampling score $M$, $g$ could sample discrete edges according to the predicted scores [55]. For two-piece graph datasets and DrugOOD datasets, we will directly use the score to reweight the messaging passing process along the edge, as we empirically find it yields more stable performance. While for CMNIST-sp and Graph-SST2, we will sample a ratio $r\%$ of all edges for each graph. The ratios adopted are 80% and 60%, respectively, following previous works [8, 28]. Meanwhile, to improve the stability of the subgraph extractor, we adopt a layernorm [66] following the practice of [55].

Besides, we also have various implementation options for obtaining the features in $\widehat{G}_c$, for further obtaining $h_{\widehat{G}_c}$, as well as for obtaining predictions based on $\widehat{G}_s$. Following previous works [55], we will adopt the same GNN encoder for the two GNNs in the interpretable GNN backbone, and feed the raw graph inputs to the classifier GNN. The contrastive loss is obtained via the graph representations of the sampled subgraph by the classifier GNN. For classifying $G$ based on $\widehat{G}_s$, we use a separate MLP downstream classifier in the classifier GNN $f_c$.

**Optimization and model selection.** By default, we use Adam optimizer [34] with a learning rate of $1e-3$ and a batch size of 128 for all models at all datasets. Except for CMNIST-sp, we use a batch size of 256 to facilitate the evaluation following previous works [55]. To avoid underfitting, we pre-train models for 20 epochs for all datasets by default. While in two-piece graphs, we find pre-training by 100 epochs yields more stable performance. To avoid overfitting, we also employ an early stopping of 5 epochs according to the validation performance. Meanwhile, dropout is also adopted for some datasets. Specifically, we use a dropout rate of 0.5 for all of the realistic graph datasets, following previous works [8, 28].

---

[4]https://drugood.github.io/

The final model is selected according to the performance at the validation set. All experiments are repeated with 5 different random seeds of $\{1, 2, 3, 4, 5\}$. The mean and standard deviation are reported from the 5 runs.

**Implementations of Euclidean OOD methods.** When implementing IRM [3], V-Rex [38] and IB-IRM [2], we refer the implementations from DomainBed [23]. Since the environment information is not available, we perform random partitions on the training data to obtain two equally large environments for these objectives following previous works [8, 12]. Moreover, we select the weights for the corresponding regularization from $\{0.01, 0.1, 1, 10, 100\}$ for these objectives according to the validation performances of IRM and stick to it for others, since we empirically observe that they perform similarly with respect to the regularization weight choice. For EIIL [13], we use the author-released implementations about assigning different samples the weights for being put in each environment and calculating the IRM loss.

**Implementations of invariant graph learning methods.** We implement GSAT [55], GREA [45], CAL [64], MoleOOD [77], GIL [41], DisC [17], and CIGA [8], according to the author provided codes (if available).

- GREA [45]: We use a penalty weight of $1$ for GREA as we empirically it does not affect the performance by changing to different weights.
  - Interpretable ratio: same as others;
  - Penalty weight: $1$;
  - Number of environments: N/A;

- GSAT [55]: We follow the recommendations of the released implementations by the authors.
  - Interpretable ratio: $70\%$;
  - Penalty weight: $1$;
  - Decay ratio: $10\%$;
  - Decay interval: `pretrain epoch`$//2$;
  - Number of environments: N/A;

- CAL [64]: We follow the recommendations of the released implementations by the authors.
  - Interpretable ratio: same as others;
  - Penalty weight: $\{0.1, 0.5, 1.0\}$;;
  - Number of environments: N/A;

- MoleOOD [77]: We tune the penalty weights of MoleOOD with values from $\{1e-2, 1e-1, 1, 10\}$ but did not observe much performance differences. Hence we stick the penalty weight as $1$ for all datasets.
  - Interpretable ratio: N/A;
  - Penalty weight: $1$;
  - Number of environments: same as others;

- GIL [41]: We follow the recommendations of the paper.
  - Interpretable ratio: same as others;
  - Penalty weight: $\{1e-5, 1e-3, 1e-1\}$;
  - Number of environments: same as others;

- DisC [17]: We tune only the $q$ weight from $\{0.9, 0.7, 0.5\}$ in the GCE loss as we did not observe performance differences by changing the weight of the other terms.
  - Interpretable ratio: same as others;
  - $q$ weight: $\{0.9, 0.7, 0.5\}$;
  - Number of environments: same as others;

- CIGA [8]: We follow the recommendations of the released implementations by the authors..
  - Interpretable ratio: same as others;
  - Penalty weight: $\{0.5, 1, 2, 4, 8, 16, 32\}$;
  - Number of environments: N/A;

- GALA:

  - Interpretable ratio: same as others;

  - Penalty weight: $\{0.5, 1, 2, 4, 8, 16, 32\}$;

  - Environment assistant: $\{\texttt{vanilla GNN}, \texttt{XGNN}\}$;

  - Sampling proxy: $\{\texttt{label predictions}, \texttt{cluster predictions}\}$;

  - Number of environments: same as others;

All of the graph learning methods adopt an interpretable GNN as the backbone by default. The only exception is MoleOOD, we follow the original implementation while using a shared GNN encoder for the variational losses to ensure the fairness of comparison. Besides, for DisC, we find the soft masking implementation in two-piece graphs will incur a severe performance degeneration hence we use a ratio of $25\%$ for the interpretable GNN backbone.

For environment inferring methods, we search the number of environments

- Two-piece graphs: fixed as 3 (since there are 3 spurious graphs);

- CMNIST-sp: 2 (since there are 2 environments);

- Graph-SST2: $\{2, 3, 4\}$ following previous practice [41];

- DrugOOD datasets: $\{2, 3, 5, 10, 20\}$ following previous practice [77].;

**Implementations of GALA.** For a fair comparison, $\texttt{GALA}$ uses the same GNN architecture for GNN encoders as the baseline methods. By default, we fix the temperature to be 1 in the contrastive loss, and merely search the penalty weight of the contrastive loss from $\{0.5, 1, 2, 4, 8, 16, 32\}$ according to the validation performances, following the $\texttt{CIGA}$ implementations [8]. By default, we implement the environment assistant as a ERM model, and adopt directly the environment assistant predictions to sample possible and negative graph pairs. Nevertheless, as discussed in Sec. 4 that there could be multiple implementation choices for the environment assistant and the use of its predictions. We hence also try with XGNN based environment assistant model and clustering based proxy predictions. By default, the selection of the environment assistant model is performed via best training performance, as which encourages a better fit to the dominant subgraph patterns, while we also try the model selection with best validation performance in DrugOOD datasets and find it empirically sometimes leads to better performance. All the options for the selection of the environment assistant models depend on the validation performance. For Two piece graphs, EC50-Scaffold, EC50-Size, Ki-Assay, Ki-Scaffold, CMNIST-sp and Graph-SST2, we find implementing the environment assistant as a ERM model already yield impressive improvements. While for the other DrugOOD datasets, we implement the environment assistant as an interpretable GNN trained with ERM and cluster the learned graph representations of the model to sample positive and negative pairs.

Since $\texttt{GALA}$ imposes a strong regularization to the data that may hinder the learning of graph representations, we pre-train the model by 10 epochs using ERM and then impose the $\texttt{GALA}$ penalty implemented as one-side contrastive loss as discussed in Sec. F. When the numbers of positive and negative pairs are extremely imbalanced, we will upsample the minor groups by a factor of $\{2, 3, 4\}$, depending on the validation performance.

### G.3   Software and Hardware

We implement our methods with PyTorch [59] and PyTorch Geometric [18]. We ran our experiments on Linux Servers installed with V100 graphics cards and CUDA 10.2.

## G.4 Computational analysis

Table 6: Averaged total training time of different methods.

| Datasets | Two-piece graphs | EC50-Assay | CMNIST-sp | Graph-SST2 |
|---|---|---|---|---|
| ERM | $435.85_{\pm2.14}$ | $80.45_{\pm10.27}$ | $315.84_{\pm5.55}$ | $374.31_{\pm1.28}$ |
| XGNN | $673.82_{\pm0.81}$ | $126.65_{\pm17.57}$ | $591.09_{\pm11.48}$ | $722.44_{\pm48.51}$ |
| GREA | $1128.28_{\pm34.57}$ | $210.30_{\pm21.23}$ | $902.06_{\pm8.49}$ | $979.15_{\pm18.3114}$ |
| GSAT | $1205.67_{\pm62.54}$ | $228.88_{\pm25.04}$ | $791.55_{\pm15.67}$ | $949.57_{\pm97.68}$ |
| DisC | $1244.68_{\pm4.76}$ | $207.50_{\pm17.72}$ | $932.40_{\pm76.99}$ | $1280.77_{\pm551.97}$ |
| MoleOOD | $714.06_{\pm6.53}$ | $136.39_{\pm17.87}$ | $439.49_{\pm9.10}$ | $712.31_{\pm81.62}$ |
| GIL | $533.46_{\pm11.42}$ | $279.30_{\pm25.39}$ | $919.53_{\pm14.15}$ | $733.36_{\pm147.08}$ |
| CIGA | $873.49_{\pm16.21}$ | $167.63_{\pm1.10}$ | $650.94_{\pm5.01}$ | $792.10_{\pm59.12}$ |
| GALA-cluster | $811.41_{\pm3.20}$ | $147.97_{\pm2.05}$ | $756.41_{\pm21.63}$ | $765.32_{\pm20.86}$ |
| GALA-pred | $793.27_{\pm8.58}$ | $149.89_{\pm2.71}$ | $644.78_{\pm53.58}$ | $764.69_{\pm30.98}$ |

We calculate the average total training time of different methods at various datasets in seconds. As shown in Table. 6, the training of GALA (no matter with clustering based sampling or prediction based sampling) does not bring much additional overhead than its counterpart CIGA. When considering the additional training time of the assistant model with ERM, GALA costs only a competitive training time as environment generation based methods such as GREA and DisC. Notably, some methods such as DisC and GIL sometimes may be slow to converge even with the same early stop setting, which will cost even more time than the time cost by GALA plus the ERM training. Besides, the ERM training time (for a assistant model) is not much long and usually around 5mins (or 300seconds in the table).

