# OpenReview forum: "Does Invariant Graph Learning via Environment Augmentation Learn Invariance?"
_NeurIPS.cc/2023/Conference — NeurIPS 2023 poster_

### Official Review · Reviewer_voKo · 2023-07-05

**Soundness:** 2 fair
**Presentation:** 2 fair
**Contribution:** 2 fair
**Rating:** 4
**Confidence:** 4

**Summary:**

The paper proposes a new framework called Graph invAriant Learning Assistant (GALA) to address the problem of learning invariance among data from different environments for out-of-distribution generalization on graphs. The authors argue that the performance of graph neural networks (GNNs) can be seriously degraded by graph distribution shifts caused by mismatches between the training and test graph distributions. GALA incorporates an assistant model to differentiate the variations in spurious subgraphs and identify the underlying invariant subgraphs. The authors conduct extensive experiments on 12 datasets with various graph distribution shifts and show that GALA brings improvements in multiple graph datasets.

**Strengths:**

1. Originality: The paper proposes a novel framework, Graph invAriant Learning Assistant (GALA), that introduces a new assistant model to differentiate the variations in spurious subgraphs and identify the underlying invariant subgraphs.
2. Quality: The authors do extensive experiments to validate the effectiveness of the proposed approach. They also provide a detailed analysis of the results and compare their approach with several state-of-the-art methods.
3. Clarity: The paper explanations of the key concepts, the methodology, and detailed experimental results.
4. Significance: The paper's contributions are significant because they provide a new perspective on the problem of learning invariant graph representations and offer a practical solution that can be applied to real-world problems.


**Weaknesses:**

1.	It would be preferable to provide the definition of the problem before presenting your proposed method.
2.	The novel concepts introduced in your proposal would benefit from a more detailed explanation. For instance, the concept of 'two-piece graphs' is somewhat confusing. It would be helpful if you could elaborate on why you have incorporated this concept and provide additional motivation.
3.	It appears that Assumption 3.5 may not appropriately be considered as an assumption. Instead, you could simply explain that $H(C|Y)\neq H(S|Y)$. Treating "either $H(C|Y)>H(S|Y)$ holds or $H(C|Y)<H(S|Y)$ holds" as an assumption may seem unusual in this context.


**Questions:**

1. I have difficulty understanding the expression $G_{c}:=f_{gen}^{G_c}(Y^{e}\cdot Rad(\alpha_{e}))$ on line 125. It seems to me that $G_{c}$ is determined purely by the label and not influenced by randomness ($Rad(\alpha_{e})$). Could you please clarify this for me?
2. In your discussion, you mentioned that when $\hat{G}_c$ contains parts of $G_s$, assigning the same labels to the generated graph is more likely to reinforce the spurious correlation between $G_s$ and $Y$. I would appreciate it if you could provide a detailed explanation of how your method effectively prevents this pitfall.
3. On line 179, you stated that "Any learning algorithm will fail in either $E_{tr}$ or $E_{tr}$". Could you please explain this sentence using clearer and more precise mathematical language?
4. In Figure 1, the last sentence mentions "pulling graphs with the same label but from $\{G_{p}\}$ and $\{G_{n}\}$ closer in the latent space." However, the figure seems to depict "pulling graphs with different labels but from $\{G_{p}\}$ and $\{G_{n}\}$ closer in the latent space." Is this a typographical error?
5. I am curious about your final loss function as mentioned in Algorithm 1. Equation 9 does not seem to be optimized directly. Could you please clarify what the final loss function is?


**Limitations:**

The writing in the paper is somewhat ambiguous, leading to occasional confusion. For more specific details, please refer to the sections labeled 'Questions' and 'Weaknesses' mentioned earlier.

---

> ### Author Rebuttal · Authors · 2023-08-09
>
> Thank you for your time in reviewing our work. Please see our detailed responses to your comments and suggestions below where we use reference numbers in our manuscript due to the character limit.
>
> > W1 Problem definition
>
> We complemented the problem definition in Sec. 2:
>
> The goal of OOD generalization on graphs is to find a GNN $f$ that minimizes the maximal loss among all environments, i.e., to minimize $\max_{e\in \mathcal{E}_\text{all}} R^e$, where $R^e$ is the risk of $f$ under environment $e$.
>
> > W2 Motivation of ``two-piece’’ graphs.
>
> We have revised our paper to include the explanation for why adopting the two-piece graphs. The main reason is that the two-piece graph is the **minimal implementation** of PIIF distribution shifts in Fig. 2, which is more suitable for theoretical discussions. Specifically, PIIF distribution shifts in Fig. 2 model the invariant and spurious correlations via the correlation of subgraphs with the labels. For simplicity, we restrict the types of invariant and spurious subgraphs and use a simple flipping operation to simulate the invariant/spurious correlations, motivated by [25] that is used to discuss the failures of IRM[3].
>
> > W3 Assumption 3.5
>
> We have revised it following your suggestion.
>
> > Q1 Understanding line 125.
>
> By definition, the Rad function will flip the labels with a probability $\alpha$. Considering a $f^{G_c}_{gen}$ that maps +1 to a ``house`` subgraph and -1 to a ``cycle`` subgraph. Let the sampled $Y^e=+1$, and $\alpha=0.1$, then with a probability of 10%, $G_c$ will be a ``cycle`` and a probability of 90%, $G_c$ will be a ``house``.
>
> > Q2 How GALA addresses the failure of environment generation?
>
> In fact, the faithfulness of environment generation **relies on the success of partitioning the invariant and spurious subgraphs**, when without additional knowledge. If the model already learns to partition the graph, then it is unnecessary to generate any new environments. Otherwise, environment generation can hardly avoid generating harmful environments for OOD generalization on graphs, as shown in Proposition 3.2.
>
> Therefore, **GALA avoids the failures of environment generation by not performing environment generation**. Instead, GALA aims to identify the maximal invariant subgraph among the partitioned subsets by the assistant model.
>
> > Q3 Explanation of Proposition 3.4.
>
> Any invariant graph learning algorithm accepts the input of the joint distribution $P(Y,G)$ and aims to identify an invariant subgraph $\widehat{G}_c$. Given the same $P(Y,G)$, the algorithm will stick to the same $\widehat{G}_c$:
>
> If $\mathcal{E}\_\text{tr}$ and $\mathcal{E}\_\text{tr}’$ have the same $P(Y,G)$ but different underlying invariant subgraphs ($G_c$, $G_c’$ respectively, with $G_c\neq G_c’$). If the algorithm succeeds in one environment, for example, $\mathcal{E}\_{tr}$, and outputs $\widehat{G}\_c=G_c$ given the input $P(Y,G)$. Then, the algorithm will fail in $\mathcal{E}\_{tr}’$ as $\widehat{G}_c=G_c\neq G_c’$.
>
> > Q4 Caption of Figure 1.
>
> We need to clarify that **the black colored labels are the predictions yielded by the assistant model instead of the ground truth labels (in orange color)**. According to the correctness of the assistant model, we can partition the whole dataset into positive graphs (i.e., correctly predicted samples in the top left and the bottom right) and negative graphs (i.e., incorrectly predicted samples in the top right and the bottom left). GALA will pull graphs with the same ground truth label but from positive and negative graphs closer.
> We revised the caption to highlight that the labels are graph labels in the figure.
>
> > Q5 The final loss.
>
> Eq. 9 is implemented in a contrastive loss. We provide the specific loss in Eq. 18 of Appendix F. We revised the text around Eq. 9 to clarify this point.
>
> Please let us know if you have any further questions. We’d be grateful if you could consider our responses above when making final evaluations of our work.

---

> ### Author Response · Authors · 2023-08-17
> **A summary of rebuttal in response to Reviewer voKo**
>
> Dear Reviewer voKo,
>
> Thank you for reviewing our paper. We have addressed your concerns, which are summarized as follows:
>
>
> - We have improved the presentation according to your suggestions, by including the definition of the problem, the explanation of the “two-piece graphs”, and making Assumption 3.5 more concise;
> - We have clarified the randomness involved in our data model and how our method addresses the previous failure case;
> - We have provided a mathematical explanation for line 179, Fig. 1, and the final loss function used in Algorithm 1.
>
> Please let us know if you have any other concerns/questions and we would try our best to address them.

---

> > ### Comment · Reviewer_voKo · 2023-08-17
> >
> > Thanks for the responses.
> >
> > I still have some concerns:
> > 1. The OOD problem studied in this paper seems to focus only on the specific case: correlation shift, but leaving the covariate shift untouched. If only correlation shift is covered, it is hard to claim the "successful OOD generalization".
> > 2. There are some related works missing, in both lines of data augmentation and invariant learning on graphs:
> >   - G-Mixup: Graph Data Augmentation for Graph Classification. ICML 2022
> >   - Causal Attention for Interpretable and Generalizable Graph Classification. KDD 2022
> >   - Graph Rationalization with Environment-based Augmentations. KDD 2022
> >   - Robust Optimization as Data Augmentation for Large-scale Graphs. CVPR 2022
> >
> > So currently, I keey my score unchanged.

---

> > > ### Author Response · Authors · 2023-08-17
> > > **Response to new questions of Reviewer voKo**
> > >
> > > Thanks for your response. Let us clarify your new questions.
> > >
> > > Regarding the distribution shifts studied in this paper, we studied the distribution shifts characterized by the causal models of [1], following the line of invariant learning [2], which could be generalized to both conditional and covariate shifts[1,3,4]. We need to clarify that our focus is to study the feasibility of learning invariant subgraphs for **successful OOD generalization under the causal model**, instead of a single type of distribution shift.
> > >
> > > We thank you for bringing up these related works that we have included the discussion of them in our revised paper. We need to clarify that:
> > > - [5,6] are general graph data augmentation methods that can be applied to most of the methods. We believe it is a promising future direction to incorporate [5,6] into the methods to further improve the OOD generalization on graphs;
> > >
> > > - [7,8] both are variants of DIR [9] that seek for generating new environments by simulating "causal intervention", i.e., by concatenating the information of invariant subgraphs with various spurious subgraphs. In Sec. 3.1, we showed that it is impossible to generate faithful new environments for boosting OOD generalization performance on graphs. **We indeed discussed and evaluated GREA[8] in our paper**, which has shown promising improvements over DIR. However, GREA still fails to achieve a satisfactory OOD generalization performance, aligned with our theory.
> > >
> > > We hope our response could clarify your new questions. Please let us know if there are any further comments/questions. We would appreciate it if you could jointly consider our responses when making the final evaluation of our paper.
> > >
> > > **References**
> > >
> > > [1] Learning causally invariant representations for out-of-distribution generalization on graphs, NeurIPS'22.
> > >
> > > [2] Invariant risk minimization, arXiv'22.
> > >
> > > [3] Invariant Models for Causal Transfer Learning, JMLR'18.
> > >
> > > [4] GOOD: A Graph Out-of-Distribution Benchmark, NeurIPS'22.
> > >
> > > [5] G-Mixup: Graph Data Augmentation for Graph Classification, ICML'22.
> > >
> > > [6] Robust Optimization as Data Augmentation for Large-scale Graphs, CVPR'22.
> > >
> > > [7] Causal Attention for Interpretable and Generalizable Graph Classification, KDD'22.
> > >
> > > [8] Graph Rationalization with Environment-based Augmentations, KDD'22.
> > >
> > > [9] Discovering invariant rationales for graph neural networks, ICLR'22.

---

> ### Author Response · Authors · 2023-08-19
> **Follow-up response**
>
> Dear Reviewer voKo,
>
> We would appreciate it very much if you could find time to take a look at our responses to your concerns, as the rebuttal will be closed in less than 2 days. We hope we would be able to discuss with you the concerns you have raised within the rebuttal period. Thank you.

---

> ### Comment · Senior_Area_Chairs · 2023-08-20
> **Please take a look at author response and let us know if your opinion has changed.**
>
> Thank you.

---

> ### Author Response · Authors · 2023-08-21
> **More results**
>
> Thank you again for your efforts in reviewing our paper. To further justify our claims and theoretical results, we additionally provide results of CAL[7] on two piece graphs. It can be found that CAL[7] has the same behaviors as GREA[8] under invariance dominated and spuriousness dominated cases.
>
> | Datasets  | {0.8, 0.6}   | {0.8, 0.7}   | {0.8, 0.9}   | {0.7, 0.9}   |
> |-----------|--------------|--------------|--------------|--------------|
> | ERM       | 77.33(±0.47) | 75.65(±1.62) | 51.37(±1.20) | 42.73(±3.82) |
> | GREA      | 82.11(±0.38) | 81.39(±0.86) | 57.38(±4.64) | 41.20(±2.99) |
> | CAL       | 82.67(±0.24) | 81.39(±0.86) | 51.87(±1.59) | 39.18(±2.39) |
> | GALA      | 81.69(±0.57) | 79.15(±0.60) | 72.03(±2.68) | 67.10(±2.44) |
>
> As the discussion phase is closing, we sincerely hope that you could take our previous responses into consideration when evaluating our paper.

---

### Official Review · Reviewer_3cig · 2023-07-06

**Soundness:** 2 fair
**Presentation:** 1 poor
**Contribution:** 3 good
**Rating:** 5
**Confidence:** 3

**Summary:**

This paper investigates OOD problems on graphs without environmental information. The authors show that it is impossible to learn invariant graph representations without additional assumptions. The authors propose a novel graph OOD method namely GALA based on a new assumption. An auxiliary model is used to make proxy predictions and then the proxy predictions are used to help identify invariant subgraphs. Experiments on 12 datasets show the effectiveness of the proposed method.

**Strengths:**

S1. The authors clearly demonstrate that the identifiability of causal subgraphs is ill-posed without environment labels. Thereby, the authors give A3.3 and A3.5 and then propose GALA based on these two assumptions.

S2. Both theoretical and experimental analyses are used to support the claim.

**Weaknesses:**

W1. This is a math-heavy paper, but the notations are unclear to me, thus making it hard to read. For example, in lines 78-79, should there be something like $e \in \mathcal{E}_{all}$? In Equation 1, there is no definition about $\widehat{G}_c$. There are more examples, I will not list them all.

W2. I think GALA can be used as a general OOD method. Why do the authors focus on graph data? I don't see any specific design only for graphs. Why not test GALA on other data types (ex., images)?

W3. I don't understand why ERM still learns the spurious correlations in invariance dominated cases. Can this be justified theoretically?

**Questions:**

Q1. I am confused about how to get the Oracle performance on these datasets. Can the authors provide more explanations?

Q2. I don't understand why Disc fails on all the datasets except CMNIST-sp. Besides, on CMNIST-sp, only GALA and Disc have some improvement over ERM.


**Limitations:**

The authors didn't discuss the limitations of the proposed method.

---

> ### Author Rebuttal · Authors · 2023-08-09
>
> Thank you for taking the time in reviewing our work. Please see our detailed responses to your comments and suggestions below where we use reference numbers in our manuscript due to the character limit.
>
> > W1 Notations
>
> We understand it might be difficult for readers who are not familiar with the literature to follow our notations and hence provide a table of key notations in Appendix A for your reference.
>
> We have revised our draft to include $e\in\mathcal{E}\_{all}$ in line 78. $\widehat{G}_c$ is the estimated subgraph, and we have added its definition before Eq 1, which is the output of the subgraph extractor.
>
> We also double checked and corrected the other parts of our draft.
>
> > W2 Why focusing on graph data?
>
> First, we need to clarify that, as pointed out in CIGA[6], OOD generalization on graphs is fundamentally more challenging than that in Euclidean data. Given that graph is a generalized form of Euclidean data (e.g., images, languages), **methods developed on graphs usually are not limited to graphs and can be applied directly to Euclidean data, but methods developed in Euclidean data can hardly be applied directly to graphs**.
>
> The specialization of GALA for graph data mainly lies in three aspects:
>
> - **The study of GALA is more desired by graph data**. Different from Euclidean data, all OOD methods on graph data [6,14,35,38,44,55,56,60,64] do not assume the availability of environment labels nor the auxiliary information [36]. Given the faithfulness of environment augmentation has never been discussed on graphs, the study of faithfulness is more desired by graph data.
> - **Graph data is also a suitable testbed for GALA**. It can be found that, on the two-piece graph datasets that satisfy the theory well, GALA outperforms other methods by a large margin up to $30$% and $13%$% in average, owing to its theoretical guarantee.
> - **GALA inherits the graph specific design from existing works** such as CIGA[6] and GSAT[44] that use information theory to handle the complexity of graph data and an interpretable GNN architecture to address graph distribution shifts modeled by the SCM in Fig. 2. Extracting the invariant subgraph is the central task specific for OOD generalization on graphs. Therefore, all of our theory and solution concentrates on the identifiability of the invariant subgraph.
>
> > W3 Why ERM still learns spurious correlations in invariance-dominated cases?
>
> It is theoretically proved in [8]. Intuitively, ERM will learn **any correlations presented in the data**, no matter whether the correlation is invariant or spurious. If we look into the first two columns in Table 1, it can be found that **GNNs trained with ERM still have a large gap from the oracle models**, meaning that GNNs trained with ERM still learn the spurious correlations.
>
> > Q1 Oracle performance
>
> The Oracle performance is obtained via canceling the distribution shifts among training and test data by randomly re-splitting the whole dataset, following [6].
>
> > Q2 Why DisC fails on all datasets except for CMNIST-sp and why only GALA and DisC succeed in CMNIST-sp?
>
> CMNIST-sp is converted from the ColoredMNIST dataset[3] which contains severe spuriousness dominated shifts in node features. DisC is designed to tackle this kind of severe distribution shifts in CMNIST-sp via an **aggressive strategy** to discard the dominant patterns [14,46], and additionally incorporate environment generation procedure based on the aggressively separated subgraphs. However, the aggressive strategy in DisC may affect learning useful patterns for generalization and fails when the distribution shifts are mild.
>
> Other methods fail in CMNIST-sp because of various drawbacks. For example, GREA, CIGA can only handle invariance dominated cases. In contrast, GALA succeeds because of its theoretical guarantee in handling both invariance and spuriousness dominated cases.
>
> > L1 Limitation discussion.
>
> We actually provided the discussion of limitations and future works in Appendix B.
>
> Please let us know if there are any further questions. We’d appreciate it if you could take the above responses into consideration when making the final evaluation of our work.

---

> > ### Comment · Reviewer_3cig · 2023-08-14
> > **Reply to authors**
> >
> > Thanks for your replies. Most of my concerns are addressed. I will increase my rating.

---

> > > ### Author Response · Authors · 2023-08-15
> > > **Thank you for reading our rebuttal**
> > >
> > > Thank you for checking our rebuttal and we're glad that we addressed more of your concerns. If any of your remaining concerns are not addressed, do please let us know and we would try our best to address them, or at least discuss with more details in the paper as we believe this could improve our work. Thank you again.

---

### Official Review · Reviewer_m4Xg · 2023-07-06

**Soundness:** 3 good
**Presentation:** 3 good
**Contribution:** 3 good
**Rating:** 6
**Confidence:** 3

**Summary:**

This paper discusses the challenges of learning invariance among data from different environments for out-of-distribution generalization on graphs. The authors propose a new framework called Graph invAriant Learning Assistant (GALA) that incorporates an assistant model to differentiate the variations in spurious subgraphs. The paper also includes experiments on 12 datasets, including DrugOOD, with various graph distribution shifts to verify the superiority of GALA.

**Strengths:**

- The paper tackles an interesting problem. Distributions shifts between training and test sets hinder and limit the potential applicability of graph neural networks methods into real world scenarios.
- To the best of my knowledge, the method proposed in this paper is novel.
- The paper is, mostly, well written and makes an effort to be easily understandable.
- The paper presents valid theoretical arguments to back its main claims.
- Finally, the paper presents an extensive sets of experiments that back the validity of the proposed method.


**Weaknesses:**

- In the tables that contain results, the best values are bolded. However, the presentation of the results would be more fair if two different colours/markers would be used when the best result is within the confidence interval of another result.

- The paper would be easier to follow if, along figure 2, a plausible real world example was used.

- In the last part of the introduction section the authors convey their contributions. In my opinion, this part is one of the less clear in the paper. To improve the clarity, I suggest the authors make a bullet list of their contributions. In particular, I would like the authors to distinguish their original contributions and how they differ from the work of Lin et al [36].


**Questions:**

I am interested in knowing if the authors believe there exists a connection between this work and the explainability literature. In particular, the counterfactual explainability subfield (like the work from [A], just to provide a reference).

_[A] Lucic, Ana, et al. "Cf-gnnexplainer: Counterfactual explanations for graph neural networks." International Conference on Artificial Intelligence and Statistics. PMLR, 2022._


**Limitations:**

-

---

> ### Author Rebuttal · Authors · 2023-08-09
>
> Thank you for your constructive suggestions and your positive feedback! Please see our detailed responses to your comments and suggestions below.
>
> > W1 Using two markers wrt. confidence interval.
>
> We revised our table and add shadows to highlight entries with the best performance and a confidence interval excluding the second best results (mean-1*std > mean of the second best result), which considers both the generalizability and stability of the best performed method.
>
> The revised table is shown in the pdf attached in the "global" response, where we can find that the improvements of GALA is non-trivial on many datasets.
>
> > W2 A plausible real word example of Fig. 2.
>
> The SCM of Fig. 2 is primarily motivated by the real-world drug property prediction task[B], where the property of a drug molecule can usually be characterized by a sub-molecule or functional group (e.g., `-OH` to solubility). We have added the discussion to the introduction of Fig.2.
>
> > W3 Listing contributions and comparing with [36]
>
> We revised our paper and included a bullet list of our contributions:
> - We identify failure cases of existing invariant graph learning approaches and establish the minimal assumptions for feasible invariant graph learning;
> - We develop a novel framework GALA with provable identifiability of the invariant subgraph for OOD generalization on graphs under the assumptions;
> - We conduct extensive experiments to verify both our theoretical results and the superiority of GALA;
>
> Notably, both our theory and solution differ from Lin et al. [36] fundamentally in terms of **assumptions** and **methodology**. More specifically, we do not rely on the auxiliary information and are compatible with the existing interpretable and generalizable GNN architecture for OOD generalization on graphs.
>
> > Q1 Connection with the explainability literature.
>
> Thanks for pointing out the related work [A]. In fact, there is a deep connection between interpretability and generalizability [6,44]. As characterized by the SCMs (Fig. 2), only extracting and using the underlying invariant subgraph for the prediction can generalize under various graph distribution shifts (Proposition E.2 in [6]), meanwhile the extracted subgraph can also serve as the interpretable result for the prediction.
>
> Both [A] and our work are inspired to incorporate causality in finding faithful interpretable subgraphs, while [A] focuses on *post-hoc explainability* given a trained GNN and ours focuses on *intrinsic interpretable GNN*. We have included the discussion of [A] in our revised draft.
>
> We’d appreciate it if you could take the above responses into consideration when making the final evaluation of and supporting our work. Please let us know if there are any outstanding questions.
>
> **References**
>
> [6] Learning causally invariant representations for out-of-distribution generalization on graphs, NeurIPS'22.
>
> [36] ZIN: When and how to learn invariance without environment partition? NeurIPS'22.
>
> [A] Cf-gnnexplainer: Counterfactual explanations for graph neural networks, AISTATS’22.
>
> [B] The rise of fragment-based drug discovery, Nature chemistry, 1(3):187, 2009.

---

> ### Author Response · Authors · 2023-08-17
> **A summary of rebuttal in response to Reviewer m4Xg**
>
> Dear Reviewer m4Xg,
>
> Thank you for reviewing our paper. We have addressed your concerns, which are summarized as follows:
>
>
> - We have revised our tables and highlighted the best results with a confidence interval excluding the corresponding second best result, as shown in the rebuttal pdf.
> - We have provided real world examples for Fig.2 for readers easier to follow and listed our contributions distinguishing from Lin et al. [36].
> - We have discussed the connection between the explainability literature (e.g., [A]).
>
>
> Please let us know if you have any other concerns/questions and we would try our best to address them.

---

> ### Author Response · Authors · 2023-08-19
> **A gentle reminder for the closing rebuttal window**
>
> Dear Reviewer m4Xg,
>
> We would appreciate it very much if you could find time to take a look at our responses to your concerns, as the rebuttal will be closed in less than 2 days. We hope we would be able to discuss with you the concerns you have raised within the rebuttal period. Thank you.

---

> > ### Comment · Reviewer_m4Xg · 2023-08-21
> >
> > I thank the reviewer for their time and effort in making the rebuttal.
> > The authors have successfully addressed some of my concerns/questions, in particular regarding the position of this paper in the current state of the art.
> > However, having also read other reviews I was influenced by some of their points, like the concerns regarding the generalization capabilities of the method.
> > All things considered, I maintain my original score.

---

### Official Review · Reviewer_waKT · 2023-07-07

**Soundness:** 3 good
**Presentation:** 4 excellent
**Contribution:** 3 good
**Rating:** 7
**Confidence:** 4

**Summary:**

Invariant graph learning methods aim to extract invariant patterns among different training environments for OOD generalization on graphs. A plethora of approaches focuses on augmenting environment information from training samples as the environment label is difficult to obtain. This work provides both theoretical and empirical analysis of the faithfulness of the existing environment augmentation methods for learning invariant graph representations. They demonstrate that it is impossible to learn invariant graph representations by environment augmentation without a set of minimal assumptions developed in the paper, including variation sufficiency and variation consistency. Based on these assumptions, they then propose GALA, Graph invAriant Learning Assistant, a new graph invariant learning framework via an assistant model providing proxy predictions to differentiate spurious variations. Experiments on 12 real-world graph OOD datasets demonstrate GALA's effectiveness.


**Strengths:**

1. This paper is very well-written and organized. The problem studying (i.e., whether the existing environment augmentation methods are faithful for learning invariant graph representations) is significant, and the motivating synthetic experiments clearly demonstrate the pitfalls of existing environment augmentation.
2. Although the variation sufficiency and variation consistency assumptions seem intuitive, this paper is the first to develop these minimal assumptions systematically and utilize them to derive the impossibility results for existing methods.
3. GALA is technically sound and the authors provide theoretical analysis to demonstrate GALA is able to identify invariant subgraph patterns.
4. Experiments are comprehensive and extensive. The results of proof-of-concept and ablation studies are consistent with the theoretical results derived in the paper, and the real-world experiments demonstrate a considerable improvement among all the baselines.
5. The authors did a great job of citing related works (in the appendix).

**Weaknesses:**

1. The environment assistant model is trained with vanilla ERM on spuriousness-dominated data to identify spurious correlations. My major concern is the vanilla ERM model should learn both spurious correlations and invariant patterns, and then how to guarantee the environment assistant model only learns environment-dependent patterns? I wonder if the authors have tried other biased ERM models as backbone and it would be great if they could elaborate more on this question.
2. In this paper, the authors divide the training sets into spuriousness-dominated and invariance-dominated sets. I wonder what if the whole training sets only contains spurious and invariance patterns distributed equally?
3. The authors did not discuss the computational complexity of GALA. I wonder if they could discuss it in detail.

**Questions:**

1. The 3 questions mentioned in the Weaknesses part.
2. Since this work is proposed to tackle OOD generalization on graphs, I wonder if the authors made some attempts on OOD generalization on Euclidean data?


**Limitations:**

The authors discussed the broader impacts in the appendix.

---

> ### Author Rebuttal · Authors · 2023-08-09
>
> Thank you for your insightful comments and your support of our work! Please see our detailed responses to your comments and suggestions below where we use reference numbers in our manuscript due to the character limit.
>
> > W1 How to guarantee the environment assistant model only learns environment dependent patterns? What about other biased ERM models?
>
> We need to clarify that **we do not intend to let the environment assistant model learn only the environment dependent patterns**. Instead, the assistant model is also allowed to learn invariant patterns. Since spurious patterns are more likely to change between environments than invariant patterns (variation sufficiency), as shown in Fig. 3a, once the assistant model learns the spurious patterns, these spurious patterns can be differentiated from invariant ones across $\{G^p\}$ and $\{G^n\}$ according to the correctness of the predictions yielded by the assistant model.
>
> In experiments, we tried two types of ERM biased models: one is the vanilla GNN and the other is the interpretable GNN, where both of them bring improvements when incorporated into GALA. Besides, we also discussed more implementation choices in Appendix F. The results show that one could implement the assistant model to better fit the spurious patterns when additional knowledge is provided, which highlights a promising future direction as discussed in Appendix B.
>
> > W2 What about training sets with equally correlated spurious and invariant patterns?
>
> Although this case is gotten rid of by the variation consistency assumption for the impossibility of identifying an invariant subgraph, results in Fig. 5 show that GALA still brings certain improvements to previous methods. We hypothesize that there exist other factors, such as the size of spurious and invariant subgraphs, affect the fitting of spurious and invariant patterns, and consider discussing more factors other than correlation strength to the design of the assistant model as well as OOD generalization on graphs as a promising future direction.
>
> > W3 Computational complexity of GALA.
>
> We have revised our manuscript to include the analysis. The additional overhead of GALA compared to a vanilla GNN lies in the training of the environment assistant model and the sampling with the assistant model.
>
> In the table below, we calculate the average total training time of different methods at various datasets in seconds. It can be found that:
> - The training of GALA (no matter with clustering based sampling or prediction based sampling) does not bring much additional overhead than its counterpart CIGA.
> - When considering the additional training time of the assistant model with ERM, GALA cost only a competitive training time as environment generation based methods such as GREA and DisC. Notably, some methods such as DisC and GIL sometimes may be slow to converge even with the same early stop setting, which will cost even more time than the time cost by GALA plus the ERM training.
> - Besides, the ERM training time (for a assistant model) is not much long and usually around **5mins** (or 300seconds in the table).
> | Training Time (seconds) | Two-piece graphs | EC50-Assay      | CMNIST-sp         | Graph-SST2       |
> |---------------|------------------|-----------------|-------------------|------------------|
> | ERM           | 435.85-+-2.14    | 80.45-+-10.27   | 315.84-+-5.55     | 374.31-+-1.28    |
> | GREA          | 1128.28-+-34.57  | 210.30-+-21.23  | 902.06-+-8.49     | 979.15-+-18.3114 |
> | GSAT          | 1205.67-+-62.54  | 228.88-+-25.04  | 791.55-+-15.67    | 949.57-+-97.68   |
> | DisC          | 1244.68-+-4.76   | 207.50-+-17.72  | 932.40-+-76.99    | 1280.77-+-551.97 |
> | MoleOOD       | 714.06-+-6.53    | 136.39-+-17.87  | 439.49-+-9.10     | 712.31-+-81.62   |
> | GIL           | 533.46-+-11.42   | 279.30-+-25.39  | 919.53-+-14.1487  | 733.36-+-147.08  |
> | CIGA          | 873.49-+-16.21   | 167.63-+-1.10   | 650.94-+-5.01     | 792.10-+-59.12   |
> | GALA-cluster  | 811.41-+-3.20    | 147.97-+-2.05   | 756.41-+-21.63    | 765.32-+-20.86   |
> | GALA-pred     | 793.27-+-8.58    | 149.89-+-2.71   | 644.78-+-53.58    | 764.69-+-30.98   |
>
> > Q2 What if applying GALA to Euclidean data?
>
> Although GALA is primarily proposed to tackle OOD generalization on graphs, it can also be applied to Euclidean data, as Euclidean data can be considered as special graphs (while methods developed in Euclidean data can hardly be generalized to graphs as shown in our experiments). For example, one may convert Euclidean data into graphs and apply GALA to it. The experiments on CMNIST-sp and Graph-SST2 show promising results of GALA in tackling the distribution shifts on Euclidean data-converted graphs. Therefore, we believe the success of GALA on graphs can be generalized to Euclidean data as well and consider it as a promising future direction.
>
> Please let us know if you have any further questions. We’d be grateful if you could take the above responses into consideration when making the final evaluation of and supporting our work!

---

> ### Author Response · Authors · 2023-08-17
> **A summary of rebuttal in response to Reviewer waKT**
>
> Dear Reviewer waKT,
>
> We want to thank you again for your insightful comments and suggestions. We have addressed your concerns, which are summarized as follows:
>
> - We have elaborated more on the curious questions of using other biased ERM architectures and whether the assistant model could learn both invariant and spurious patterns (under any cases);
> - We have discussed the computational complexity and compared our method with other competitors;
> - We have shown that our method indeed is able to generalize to Euclidean data, as shown in the experiments on CMNIST-sp and Graph-SST2.
>
> Please let us know if you have any other concerns/questions and we would try our best to address them.

---

> ### Author Response · Authors · 2023-08-19
> **A gentle reminder for the closing rebuttal window**
>
> Dear Reviewer waKT,
>
> We would appreciate it very much if you could find time to take a look at our responses to your concerns, as the rebuttal will be closed in less than 2 days. We hope we would be able to discuss with you the concerns you have raised within the rebuttal period. Thank you.

---

> > ### Comment · Reviewer_waKT · 2023-08-19
> > **Re: Rebuttal by Authors**
> >
> > Thank you for the detailed rebuttal. I appreciate the authors providing an empirical analysis regarding GALA's computational complexity, which addresses my concern about the model's computational complexity. Regarding the questions about the environment assistant model, I now understand the intuition and agree it would learn spurious correlations on spuriousness-dominated graphs. However, I am still not fully convinced about the environment assistant model's ability to distinguish different environments, as the goal of the environment assistant model is trying to distinguish different environments from extracted spurious correlations, while the ERM-based model (e.g., vanilla GNN) learns both invariant and spurious correlations. I think the current design of the environment assistant model is in the middle of an invariant learning model and a model only learns spurious correlations. Thus, I feel maybe an extra feature-selection step or designing a biased ERM model as backbone rather than vanilla GNN might help. That being said, I think the above concerns are minor, and I would like to keep my score as an Accept.

---

> > > ### Author Response · Authors · 2023-08-20
> > > **Thank you for checking our rebuttal and for your further comments**
> > >
> > > Thank you for checking our rebuttal and for your further comments.
> > >
> > > > Regarding the environment assistant model's ability to distinguish different environments
> > >
> > > In fact, the current design of the environment assistant model aims to distinguish the variations of spurious features instead of the specific environment partitions. As spurious correlations are more unstable than invariant correlations (Variation sufficiency assumption), the partitioned positive and negative graph sets could amplify the instability of spurious correlations. Then, GALA could locate the invariant subgraphs by finding the subgraphs that preserve the maximal invariance across the positive and negative graph sets. As shown in Fig. 3a, no matter which of the invariant or spurious correlations is stronger (Variation consistency assumption), the invariant subgraph maintains the maximally invariant correlations across the positive and negative graph sets partitioned by a simple ERM-based model.
> > >
> > > Nevertheless, we agree there are limitations in the current design of the environment assistant model as you pointed out. Incorporating additional design such as selecting/identifying spurious features, or an even more biased assistant model, is definitely a promising future direction to extend and improve our framework. We have included the discussion in our revised draft.
> > >
> > > Thank you again for your constructive comments and suggestions!

---

### Official Review · Reviewer_W1GV · 2023-07-07

**Soundness:** 3 good
**Presentation:** 2 fair
**Contribution:** 3 good
**Rating:** 5
**Confidence:** 1

**Summary:**

The paper proposes Graph invAriant Learning Assistant (GALA), targeting at out-of-distribution (OOD) generalization on graphs leveraging invariant graph representation learning. Authors argue that environment augmentation cannot achieve this without additional assumptions, leading to the development of minimal assumptions—variation sufficiency and variation consistency. GALA utilizes an assistant model to distinguish variations in spurious subgraphs sensitive to environment changes or distribution shifts. Empirical experimental results support the efficacy of GALA.

**Post Rebuttal**: I have read the rebuttal of the authors and will keep my rating.

**Strengths:**

- This paper provides a thorough theoretical explanation with step-by-step illustration. It builds a strong theoretical case for the proposed framework, establishing a set of minimal assumptions required for invariant graph learning. Practically GALA has a new design using an assistant model to differentiate variations in spurious subgraphs.

- Empirically, authors present extensive experimental results on both synthetic and real-world data, showing the effectiveness of the proposed method.

**Weaknesses:**

- Considering that scalability has always been an issue for GNNs, it would be better if the authors can also discuss the overhead of further introducing a new assistant module.
- The proposed method seems specialized in the OOD generalization of graph structures, whereas the feature OOD is also a major problem in the graph OOD setting. Considering this the applicability of this method may be restrained.

**Questions:**

-  The assistant model that needs to be sensitive to graph environment changes, but how do you guarantee that?

**Limitations:**

The authors may not explicitly discuss the limitations of the proposed method.

---

> ### Author Rebuttal · Authors · 2023-08-09
>
> Thank you for your time in reviewing our paper and your positive feedback! We hope our response below would make you more confident about the evaluation of our work.
>
> > W1 Computational overhead.
>
> We have revised our manuscript to include the analysis. The additional overhead of GALA compared to a vanilla GNN lies in the training of the environment assistant model and the sampling with the assistant model.
>
> In the table below, we calculate the average total training time of different methods at various datasets in seconds. It can be found that:
> - The training of GALA (no matter with clustering based sampling or prediction based sampling) does not bring much additional overhead than its counterpart CIGA.
> - When considering the additional training time of the assistant model with ERM, GALA cost only a competitive training time as environment generation based methods such as GREA and DisC. Notably, some methods such as DisC and GIL sometimes may be slow to converge even with the same early stop setting, which will cost even more time than the time cost by GALA plus the ERM training.
> - Besides, the ERM training time (for a assistant model) is not much long and usually around **5mins** (or 300seconds in the table).
>
> | Training Time (seconds) | Two-piece graphs | EC50-Assay      | CMNIST-sp         | Graph-SST2       |
> |---------------|------------------|-----------------|-------------------|------------------|
> | ERM           | 435.85-+-2.14    | 80.45-+-10.27   | 315.84-+-5.55     | 374.31-+-1.28    |
> | GREA          | 1128.28-+-34.57  | 210.30-+-21.23  | 902.06-+-8.49     | 979.15-+-18.3114 |
> | GSAT          | 1205.67-+-62.54  | 228.88-+-25.04  | 791.55-+-15.67    | 949.57-+-97.68   |
> | DisC          | 1244.68-+-4.76   | 207.50-+-17.72  | 932.40-+-76.99    | 1280.77-+-551.97 |
> | MoleOOD       | 714.06-+-6.53    | 136.39-+-17.87  | 439.49-+-9.10     | 712.31-+-81.62   |
> | GIL           | 533.46-+-11.42   | 279.30-+-25.39  | 919.53-+-14.1487  | 733.36-+-147.08  |
> | CIGA          | 873.49-+-16.21   | 167.63-+-1.10   | 650.94-+-5.01     | 792.10-+-59.12   |
> | GALA-cluster  | 811.41-+-3.20    | 147.97-+-2.05   | 756.41-+-21.63    | 765.32-+-20.86   |
> | GALA-pred     | 793.27-+-8.58    | 149.89-+-2.71   | 644.78-+-53.58    | 764.69-+-30.98   |
>
> > W2 OOD generalization on node features.
>
> We need to clarify that, despite our discussion of failure cases focusing on graph structures, GALA works for distribution shifts on both graph structures and features, as demonstrated in experiments. For example, the CMNIST-sp dataset is converted from colored MNIST dataset where the colors have a spurious correlation with labels, and hence CMNIST-sp contains distribution shifts in node features.
>
> We are happy to answer if there are any outstanding questions, and we’d be grateful if you could jointly consider our responses above when making a final evaluation of our work.

---

> ### Author Response · Authors · 2023-08-17
> **A summary of rebuttal in response to Reviewer W1GV**
>
> Dear Reviewer W1GV,
>
> Thank you for reviewing our paper. We have addressed your concerns, which are summarized as follows:
>
> - We have discussed the computational overhead of our method and compared it with the competitors;
> - We have clarified that our method indeed can be applied to address distribution shifts on node features, as shown in CMNIST-sp experiments.
> - The sensitivity to graph environment changes of the assistant model is **guaranteed by the nature of ERM** (We provided more details in the [response](https://openreview.net/forum?id=EqpR9Vtt13&noteId=DrbfXyaIu5) to Reviewer 3cig). It is theoretically proved in [8] that ERM will learn **any correlations presented in the data**, no matter whether the correlation is invariant or spurious. If we look into the first two columns in Table 1, it can be found that **GNNs trained with ERM still have a large gap from the oracle models**, meaning that GNNs trained with ERM still learn the spurious correlations.
>
> Please let us know if you have any other concerns/questions and we would try our best to address them.

---

> ### Author Response · Authors · 2023-08-19
> **A gentle reminder for the closing rebuttal window**
>
> Dear Reviewer W1GV,
>
> We would appreciate it very much if you could find time to take a look at our responses to your concerns, as the rebuttal will be closed in less than 2 days. We hope we would be able to discuss with you the concerns you have raised within the rebuttal period. Thank you.

---

### Author Rebuttal · Authors · 2023-08-09

Dear reviewers,


Thank you for your comments and suggestions. To summarize, all reviewers agree that our theoretical analysis is clear, significant, and interesting. The motivated solution GALA is novel (waKT, m4Xg, voKo, voKo). The empirical study is comprehensive and extensive to demonstrate the superiority of GALA  (W1GV, waKT, m4Xg, voKo). The presentation is clear, well-written and easily understandable (waKT, m4Xg, voKo).

Regarding the reviewers’ concerns, we believe they can be addressed and we have revised our draft according to the reviewers’ valuable suggestions.  In the following, we give our responses to the reviewers’ concerns and suggestions. We will use reference numbers in our manuscript due to the character limit:

Regarding the specialization of GALA for graph data:
- First, we clarify that, as pointed out in CIGA[6], OOD generalization on graphs is fundamentally more challenging than that in Euclidean data. Given that graph is a generalized form of Euclidean data (e.g., images, languages), **methods developed on graphs usually are not limited to graphs and can be applied directly to Euclidean data, but methods developed in Euclidean data can hardly be applied directly to graphs**.
- The necessity and specialization of GALA for graph data originate from the following three aspects:
   - The study of GALA is more desired by graph data. Different from Euclidean data, all OOD methods on graph data [6,14,35,38,44,55,56,60,64] do not assume the availability of environment labels nor the auxiliary information [36]. **Given the faithfulness of environment augmentation has never been discussed on graphs, the study of faithfulness is more desired by graph data**.
   - **Graph data is also a suitable testbed for GALA**. It can be found that, on the two-piece graph datasets that satisfy the theory well, GALA outperforms other methods by a large margin up to $30$% and $13%$% in average, owing to its theoretical guarantee.
  - **GALA inherits the graph specific design from existing works** such as CIGA[6] and GSAT[44] that use information theory to handle the complexity of graph data and an interpretable GNN architecture to address graph distribution shifts modeled by the SCM in Fig. 2. Extracting the invariant subgraph is the central task specific for OOD generalization on graphs. Therefore, all of our theory and solution concentrates on the identifiability of the invariant subgraph.

Regarding the computational overhead, we calculated the average total training time of different methods at various datasets in seconds. It can be found that:
- The training of GALA (no matter with clustering based sampling or prediction based sampling) does not bring much additional overhead than its counterpart CIGA.
- When considering the additional training time of the assistant model with ERM, GALA cost only a competitive training time as environment generation based methods such as GREA and DisC. Notably, some methods such as DisC and GIL sometimes may be slow to converge even with the same early stop setting, which will cost even more time than the time cost by GALA plus the ERM training.
- Besides, the ERM training time (for a assistant model) is not much long and usually around **5mins** (or 300seconds in the table).
| Training Time (seconds) | Two-piece graphs | EC50-Assay      | CMNIST-sp         | Graph-SST2       |
|---------------|------------------|-----------------|-------------------|------------------|
| ERM           | 435.85-+-2.14    | 80.45-+-10.27   | 315.84-+-5.55     | 374.31-+-1.28    |
| GREA          | 1128.28-+-34.57  | 210.30-+-21.23  | 902.06-+-8.49     | 979.15-+-18.3114 |
| GSAT          | 1205.67-+-62.54  | 228.88-+-25.04  | 791.55-+-15.67    | 949.57-+-97.68   |
| DisC          | 1244.68-+-4.76   | 207.50-+-17.72  | 932.40-+-76.99    | 1280.77-+-551.97 |
| MoleOOD       | 714.06-+-6.53    | 136.39-+-17.87  | 439.49-+-9.10     | 712.31-+-81.62   |
| GIL           | 533.46-+-11.42   | 279.30-+-25.39  | 919.53-+-14.1487  | 733.36-+-147.08  |
| CIGA          | 873.49-+-16.21   | 167.63-+-1.10   | 650.94-+-5.01     | 792.10-+-59.12   |
| GALA-cluster  | 811.41-+-3.20    | 147.97-+-2.05   | 756.41-+-21.63    | 765.32-+-20.86   |
| GALA-pred     | 793.27-+-8.58    | 149.89-+-2.71   | 644.78-+-53.58    | 764.69-+-30.98   |


We have also revised our manuscript and improved its clarity thanks to the suggestions of all reviewers.

In addition, we have provided a link to our codes for reproducing the results in our paper to the AC.

We hope our previous responses could address the concerns of all reviewers. Otherwise, we are happy to discuss and provide further evidence for any outstanding questions. We would appreciate it if you could take our responses into consideration when making the final evaluation of our work.

Sincerely,
Authors

---

### Decision · Program_Chairs · 2023-09-21

**Decision:**

Accept (poster)

**Comment:**

The paper introduces the Graph invAriant Learning Assistant (GALA), a novel framework for learning invariant graph representations. The paper proves the limitation of existing environment augmentation, and establish minimal assumptions—variation sufficiency and variation consistency—for invariant graph learning. They introduce an assistant model to distinguish variations in spurious subgraphs, thereby enhancing robustness against environmental shifts.  Empirical results, both synthetic and real-world, convincingly validate the efficacy of GALA. The paper is well-structured, clear, and supported by comprehensive experiments and theoretical analyses. Overall, this work advances the understanding of OOD in graph neural networks and should be accepted.